# MultiLoReFT: Decoupling Shared and Modality-Specific Subspaces in Multimodal Learning via Low-Rank Representation Fine-Tuning

**Sana Tonekaboni** [1 2 3]   **Viktoria Schuster** [1 2 4]   **Caroline Uhler** [1 2]

## Abstract

Real-world perception and decision making are inherently multimodal, integrating complementary signals across modalities. However, training multimodal models faces two main obstacles. First, collecting large-scale, well-aligned paired multimodal datasets is often impractical, making end-to-end multimodal training difficult. Second, existing multimodal representations frequently entangle information shared across modalities with modality-specific information, hindering interpretability and control. We introduce MultiLoReFT, an efficient and scalable low-rank representation fine-tuning framework for multimodal learning with pretrained unimodal models. MultiLoReFT extends low-rank adaptation to the multimodal setting and learns interpretable projection subspaces that decouple shared and modality-specific information. Across simulated and real-world benchmarks, it produces representations that support multimodal prediction while explicitly revealing how shared and modality-specific information is distributed across modalities.[1]

## 1. Introduction

The growth of multimodal data, ranging from image-caption pairs to multimodal diagnostic data, has enabled a wide range of applications (Liang et al., 2024) in vision-and-language modeling (Chen et al., 2024; Sun et al., 2025), medical diagnostics (Steyaert et al., 2023; Zhou et al., 2023; Tonekaboni et al., 2025; Radhakrishnan et al., 2023), and biology (Cui et al., 2025; Uhler & Shivashankar, 2022; Yang et al., 2021; Zhang et al., 2022). These applications require learning effective joint representations that capture both the common semantics across modalities and the unique information each modality provides. However, training multimodal models from scratch typically demands large amounts of aligned multimodal data, which are often scarce or expensive to obtain in real-world settings like healthcare, where data collection is done in observational settings (Baltrušaitis et al., 2019). This has motivated recent work to leverage powerful unimodal encoders pretrained on large-scale single-modality data and adapt them for few-shot multimodal learning (Alayrac et al., 2022; Kim & Kim, 2024; Miyazawa et al., 2022).

While fine-tuning enables flexible reuse of pretrained models, it can be computationally intensive and parameter inefficient. Parameter-efficient fine-tuning methods such as low-rank adaptation (LoRA) (Hu et al., 2021) and low-rank representation fine-tuning (LoReFT) (Wu et al., 2024) provide efficient alternatives to full fine-tuning by learning low-dimensional updates, to selected weight or intermediate representations, while keeping most parameters frozen. These approaches achieve comparable performance to full fine-tuning with less computation (Ding et al., 2023). They are particularly attractive in data-limited regimes, such as multimodal cohorts, as they reduce the risk of overfitting while preserving pretrained knowledge.

In this work, we extend low-rank representation fine-tuning to the multimodal setting. We propose MultiLoReFT, a framework that efficiently fuses information from multiple pretrained unimodal encoders while simultaneously disentangling shared and unique information from each modality. This factorization enhances interpretability by exposing the relative contributions of each modality, facilitates the learning of cross-modal interactions, and improves the robustness to missing modalities. MultiLoReFT offers a self-supervised solution to augment unimodal representations with cross-modal information that generalize to any downstream task. It learns structured low-rank projection matrices that define orthogonal shared and modality-specific subspaces to decouple the unique information contribution of each modality. We further introduce an adaptive pruning mechanism that dynamically adjusts the rank of each pro-

---

[1]Massachusetts Institute of Technology, Cambridge, MA, USA [2]Eric and Wendy Schmidt Center, Broad Institute of MIT and Harvard, Cambridge, MA, USA [3]Vector Institute, Toronto, Canada [4]Technical University of Denmark, Lyngby, Denmark. Correspondence to: Sana Tonekaboni <stonekab@mit.edu>.

*Proceedings of the 43$^{rd}$ International Conference on Machine Learning*, Seoul, South Korea. PMLR 306, 2026. Copyright 2026 by the author(s).

[1]Code available at: https://github.com/sanatonek/MultiLoReFT

jection based on its information content, yielding a compact parameterization that avoids unnecessary redundancy while maintaining performance. We evaluate our approach on synthetic and real-world multimodal datasets, demonstrating its ability to learn shared and modality-specific subspaces that disentangle unique and shared information, produce effective multimodal representations for downstream tasks, and capture cross-modal interactions that improve robustness to missing modalities. MultiLoReFT offers a lightweight approach to leverage pretrained unimodal models in multimodal scenarios that lays the groundwork for interpretable and adaptable multimodal systems in data-scarce settings.

## 2. Related work

**Multimodal Representation Learning.** A central challenge in multimodal learning is how to integrate heterogeneous signals into effective representations (Liang et al., 2021). Early approaches rely on simple early, intermediate, or late fusion mechanisms (Boulahia et al., 2021). Coordinated representation approaches align unimodal encoders into a shared embedding space, often with contrastive or retrieval-based objectives (Radford et al., 2021). Fusion-based models are widely used, ranging from simple concatenation or pooling strategies (Baltrušaitis et al., 2019) to attention-based architectures that explicitly capture cross-modal interactions (Tsai et al., 2019; Jayakumar et al., 2020). Recent studies have taken a more analytical view, quantifying redundant and complementary information between modalities using Partial Information Decomposition (PID) (Liang et al., 2023a; Zhang et al., 2025) or Causal Representation Learning (Sturma et al., 2023). These insights emphasize that naive fusion lacks clarity on the structure of modality-specific and shared information, motivating the development of disentanglement frameworks.

**Disentangled Multimodal Representations.** Disentangling the underlying generative factors can improve interpretability by making latent representations more aligned with distinct, meaningful sources of variation in the data (Zhu et al., 2021; Makelov et al., 2025; Tonekaboni et al., 2022). Hence, a complementary line of work aims to explicitly separate shared and modality-specific information in multimodal settings. These methods often define information components with respect to downstream tasks; for instance, FactorCL (Liang et al., 2023b) aligns modality-invariant features with supervision signals while preserving unique factors using a factorized contrastive learning. Triple Disentanglement (Zhou et al., 2025) further decomposes representations into shared, relevant, and irrelevant modality-specific components using a transformer-based encoder–fusion design. Other approaches offer self-supervised alternatives to information decomposition (Wang et al., 2025). Multimodal VAE-based approaches disentangle multimodal latent spaces through variational inference (Lee &

Pavlovic, 2021; Meo & Lanillos, 2021). However, these methods often face challenges in achieving a clean separation of shared and unique information. APOLLO (Zhang et al., 2026) proposes a related alternative based on latent optimization, learning shared embeddings that are later generalized through trained encoders. Other methods like DRIM-U (Robinet et al., 2024) enforce disentanglement through reconstruction and adversarial regularization. These approaches move beyond fusion to provide a more structured account of modality interactions.

**Multimodal Fine-tuning.** The difficulty of collecting large-scale paired multimodal datasets has motivated recent work to leverage unimodal models for multimodal learning (Zhang et al., 2024). Existing approaches range from fusion of unimodal encoders (Miyazawa et al., 2022; Norelli et al., 2023) to direct fine-tuning for multimodal tasks (Zhai et al., 2022). Yet, fine-tuning large models remains challenging, particularly when available cohorts are small (Vieira et al., 2024). For large language models, parameter-efficient fine-tuning rather than full model adaptation has shown strong performance for downstream tasks (Wu et al., 2024; Hu et al., 2021). Extensions to multimodality (Liu et al., 2025) similarly demonstrate gains in both performance and flexibility. Building on this, our method introduces an efficient unsupervised multimodal fine-tuning that improves both multimodal performance and interpretability.

## 3. Multimodal Low-rank Representation Finetuning (MultiLoReFT)

We introduce a low-rank representation fine-tuning framework for multimodal learning called MultiLoReFT that decomposes pretrained unimodal representations into shared and modality-specific components. As shown in Figure 1, our method operates on top of frozen pretrained encoders, requiring only a small number of additional parameters. This design enables efficient multimodal fusion while adding interpretability by explicitly disentangling modality-specific contributions from shared information.

### 3.1. Representation Fine-tuning with MultiLoReFT

We build on representation fine-tuning (Wu et al., 2024) to adapt pretrained unimodal encoders for multimodal learning. The key idea is to apply structured low-rank interventions directly to pretrained representations to steer them into disentangled multimodal subspaces that separate shared and modality-specific information. Consider two pretrained unimodal encoders $E_1$ and $E_2$ for modalities 1 and 2. Given inputs $x_1$ and $x_2$, these encoders produce representations $\mathbf{h}_1, \mathbf{h}_2 \in \mathbb{R}^d$. We learn fine-tuned representations $\Phi_1, \Phi_2 \in \mathbb{R}^d$ that (i) preserve the expressive power of the representations, (ii) capture multimodal interactions, and (iii) disentangle shared and modality-specific information.

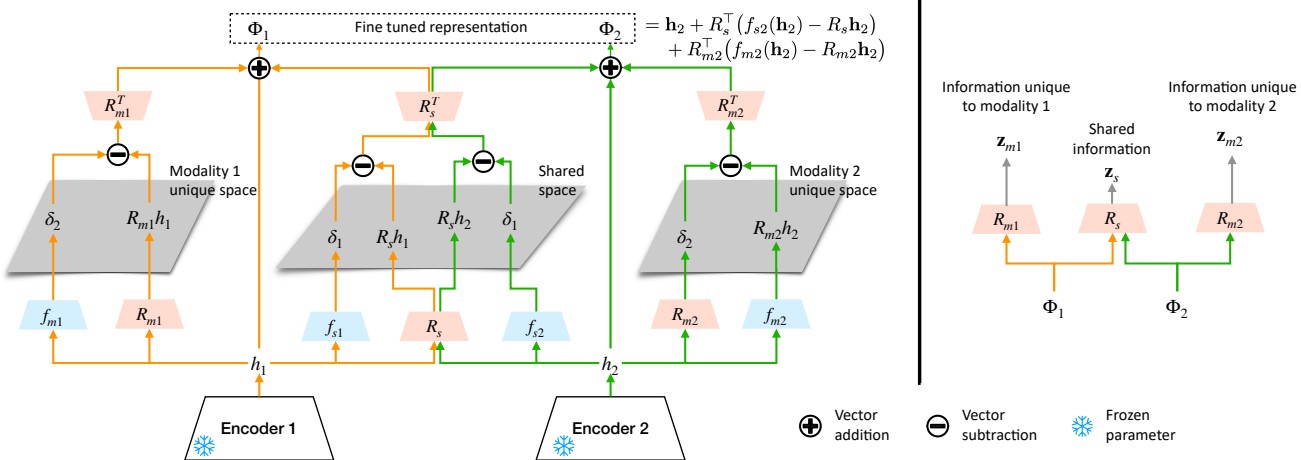

*Figure 1.* Overview of **MultiLoReFT**. Pretrained unimodal encoders (with frozen parameters) produce representations $h_1$ and $h_2$, which are fine-tuned through low-rank projections into shared ($R_s$) and modality-specific ($R_{m1}, R_{m2}$) subspaces. Nonlinear transforms ($f_{s1}, f_{s2}, f_{m1}, f_{m2}$) learn the representation edit in the lower-rank space that needs to be subtracted from the projection of the representations. Once edited in the low-rank space, the representations are projected back to form the fine-tuned representations $\Phi_1$ and $\Phi_2$. The right panel illustrates the decoupling of information into shared ($\mathbf{z}_s$) and modality-unique ($\mathbf{z}_{m1}, \mathbf{z}_{m2}$) components. $\mathbf{z}_s$ is computed as the mean of the shared projections across available modalities, using both when present and a single projection when only one is available.

MultiLoReFT defines distinct low-rank subspaces for shared and modality-specific information, where low-rank updates are applied. The unimodal representations projected into these subspaces via low-rank matrices $R_s, R_{m1}, R_{m2} \in \mathbb{R}^{r \times d}$ (Eq. 1), yield the decoupled shared and modality-specific components. $\mathbf{z}_{s1}$ and $\mathbf{z}_{s2}$ represent the shared information extracted from each modality, and $\mathbf{z}_{m1}$ and $\mathbf{z}_{m2}$ represent the unique information to each modality.

$$\mathbf{z}_{s1} = R_s \Phi_1, \quad \mathbf{z}_{s2} = R_s \Phi_2. \tag{1}$$

$$\mathbf{z}_{m1} = R_{m1} \Phi_1, \quad \mathbf{z}_{m2} = R_{m2} \Phi_2. \tag{2}$$

MultiLoReFT learns lightweight transformations $f_{si}, f_{mi}$ in each low-rank subspaces to fine-tune the unimodal representations of modality $i \in \{1, 2\}$ as shown in Eq. 3 and 4. These edits operate only in the low-rank subspaces, making training efficient while ensuring interpretability. The shared subspace isolates common information, and the modality-specific subspaces capture unique signals. The transformations learn how to update the representations in each space.

$$\Phi_1 = \mathbf{h}_1 + R_s^\top (f_{s1}(\mathbf{h}_1) - R_s \mathbf{h}_1) + R_{m1}^\top (f_{m1}(\mathbf{h}_1) - R_{m1} \mathbf{h}_1) \tag{3}$$

$$\Phi_2 = \mathbf{h}_2 + R_s^\top (f_{s2}(\mathbf{h}_2) - R_s \mathbf{h}_2) + R_{m2}^\top (f_{m2}(\mathbf{h}_2) - R_{m2} \mathbf{h}_2) \tag{4}$$

### 3.2. MultiLoReFT Learning Objective

Unlike typical parameter-efficient fine-tuning methods, MultiLoReFT training is not guided by supervised task labels but instead by structural constraints imposed on the representation space. The goal is to adapt pretrained unimodal embeddings so that their projections into learned subspaces exhibit disentanglement while still enabling effective multimodal fusion. To this end, we optimize a composite objective that encourages (i) independence between shared and modality-specific components, (ii) orthogonality between subspaces, and (iii) preservation of information from the original unimodal embeddings. The overall loss is composed of 3 components:

$$\mathcal{L} = \lambda_1 \mathcal{L}_{\text{indep}} + \lambda_2 \mathcal{L}_{\text{orth}} + \lambda_3 \mathcal{L}_{\text{MI}}. \tag{5}$$

**Independence loss. ($\mathcal{L}_{\text{indep}}$)** To ensure that shared and modality-specific components capture complementary information, we minimize their statistical dependence using the Hilbert–Schmidt Independence Criterion (HSIC) (Gretton et al., 2007). HSIC is a nonparametric measure of statistical dependence between two random variables. Given random variables $X$ and $Y$ with characteristic kernels $K$ and $L$ (e.g., Gaussian RBF or Laplace), $\text{HSIC}(X, Y) = 0$ if and only if $X$ and $Y$ are statistically independent. We use an empirical, unbiased estimator of HSIC ($\frac{1}{(n-1)^2} \text{tr}(KL)$) that can be minimized during training to enforce nonlinear independence between the representation components. $K, L \in \mathbb{R}^{n \times n}$ are centered RBF kernel matrices in our experiments. Independence is enforced between the shared and private subspaces of each modality as well as the two modality-specific subspaces to prevent leakage of redundant shared information into the private components:

$$\mathcal{L}_{\text{indep}} = \text{HSIC}(\mathbf{z}_{s1}, \mathbf{z}_{m1}) + \text{HSIC}(\mathbf{z}_{s2}, \mathbf{z}_{m2}) + \text{HSIC}(\mathbf{z}_{m1}, \mathbf{z}_{m2}). \tag{6}$$

**Orthogonality loss. ($\mathcal{L}_{\mathbf{orth}}$)**  While independence ensures statistical separation, we further enforce disjointness between the subspaces by minimizing the Frobenius norm of the pairwise inner product or the projection matrices $R_s$, $R_{m1}$, and $R_{m2}$:

$$\mathcal{L}_{\text{orth}} = \|R_s R_{m1}^\top\|_F + \|R_s R_{m2}^\top\|_F. \quad (7)$$

This constraint strengthens disentanglement by ensuring the subspaces are orthogonal. While HSIC guarantees that subspaces do not carry redundant information, orthogonality ensures that their basis vectors do not overlap in representation space. Relying on only one is insufficient, as for example, statistically independent factors can still be geometrically aligned (e.g., colinear directions in the embedding space), while orthogonal directions may still exhibit nonlinear dependence.

**Cross-modal mutual information loss. ($\mathcal{L}_{\mathbf{MI}}$)**  To ensure that the fine-tuned projections retain information from the original unimodal embeddings $\mathbf{h}$ and learn cross-modal information, we adopt an InfoNCE-style contrastive loss as shown in Equation 8. InfoNCE provides a lower bound on the true mutual information, and maximizing this bound reserves high mutual information between the projections and their sources. Therefore, the disentangled shared and modality-specific projections remain sufficient summaries of their original embeddings.

$$\mathcal{L}_{\text{MI}} = -\frac{1}{2} \sum_{i=1}^{2} \log \frac{\exp\left(\langle \mathbf{h}_i, \mathbf{z}^{(i)} \rangle / \tau\right)}{\sum_{j=1}^{N} \exp\left(\langle \mathbf{h}_i, \mathbf{z}^{(j)} \rangle / \tau\right)}. \quad (8)$$

Here, the parameter $\tau$ is the temperature parameter that controls the sharpness of the similarity distribution inside the softmax, $h_i$ is the pretrained embedding of modality $i$, and $N$ is the batch size. $\mathbf{z}^{(i)}$ is formed by concatenating the modality-specific projection from modality $i$ with the shared projection from the opposite modality. This design enforces consistency of shared components across modalities, ensuring that they encode modality-agnostic information.

We note that the representations entering the MI loss may not necessarily have the same dimensionality. To compute the MI objective consistently, we project both $h_i$ and $\mathbf{z}^{(i)}$ into a common space using a fixed, non-trainable random projection head, with dimension set to the maximum dimension of the 2 representations. The MI loss and inner product is then computed in this shared space after normalization.

### 3.3. Rank Adaptation via Pruning

A key challenge in disentangled representation learning is determining the dimensionality of shared and modality-specific subspaces. Fixing ranks a priori risks underfitting when too small, or redundancy and leakage when too large.

To address this, we adopt a dynamic rank adaptation mechanism that prunes low-energy directions (Hu et al., 2021). During training, we compute the singular value decomposition (SVD) of each projection matrix $R_s, R_{m1}, R_{m2}$ as $R = USV^\top$, with singular values $S = \text{diag}(\sigma_1, \ldots, \sigma_r)$. Dimensions with $\sigma_i$ below a threshold $\epsilon$ are pruned and the matrices are updated with a rotated, compressed basis $\tilde{R} = \text{diag}(S_{1:k}) V_{1:k}^\top$, ensuring orthogonality and alignment with dominant directions (Algorithm 1). We cap pruning to 10% of total rank in each step to allow controlled recovery. At pruning, the last layer of the transform functions are also pruned to the same dimension. Rank adaptation eliminates the need for manual rank tuning, improves disentanglement, and improves efficiency by reducing the projection ranks.

---

**Algorithm 1** Adaptive Rank Pruning

---

**Require:** Threshold $\epsilon$; matrices $R_s, R_{m1}, R_{m2}$; transforms $f_{s1}, f_{s2}, f_{m1}, f_{m2}$
1: **for all** $R \in \{R_s, R_{m1}, R_{m2}\}$ **do**
2:     Compute SVD: $R = USV^\top$, where $S = \text{diag}(\sigma_1, \ldots, \sigma_r)$ and $\sigma_1 \geq \cdots \geq \sigma_r$
3:     $\mathcal{I} \leftarrow \{i \in [r] : \sigma_i < \epsilon\}$
4:     $n_{\text{prune}} \leftarrow \min(|\mathcal{I}|, \lfloor 0.1 \cdot r \rfloor)$
5:     $k \leftarrow r - n_{\text{prune}}$
6:     Form $\tilde{R} \leftarrow \text{diag}(\sigma_1, \ldots, \sigma_k) V_{1:k}^\top$
7:     **for all** $f$ associated with $R$ **do**
8:         Replace the last linear layer of $f$ with a $k$-dimensional layer and rotate its weights by $U_{:,1:k}^\top$
9:     **end for**
10:    $R \leftarrow \tilde{R}$
11: **end for**

---

### 3.4. Training Procedure

The details of MultiLoReFT training procedure are outlined in Algorithm 2. We learn the parameters of the projection matrices $R_{m1}, R_{m2}, R_s$ as well as the transform parameters $f_{s1}, f_{m1}, f_{s2}, f_{m2}$ using the objective function in Equation 5. To avoid hand-tuning regularization weights $\lambda$, we adopt *Gradient Normalization* (Chen et al., 2018), which balances the contributions of each objective by equalizing their gradient magnitudes. We investigate the impact of gradient normalization and each loss term in an ablation study (Appendix A.2), showing that all components are necessary to obtain well-decoupled, high-quality representations.

As shown in Algorithm 2, after an initial warm-up phase of 50 epochs, adaptive rank pruning is enabled to refine shared and modality-specific subspace ranks. At the end of each subsequent epoch, the mutual information loss $\mathcal{L}_{\text{MI}}$ on the validation set is monitored, and a pruning step is triggered only if this loss remains within 1% of its best observed value, ensuring that rank reduction does not substantially degrade representation quality. Training then continues, whether or

not pruning occurs in a given epoch, until convergence. We define convergence in our experiments as a minimum improvement of 0.001 in total validation loss within a patience window of 150 epochs.

---

**Algorithm 2** MultiLoReFT Training

---

**Require:** Multimodal datasets $(\mathcal{D}_{\text{train}}, \mathcal{D}_{\text{val}})$; pretrained unimodal encoders $(E_1, E_2)$; pruning threshold $\epsilon$

1: **Variables:** Projection matrices $R_s, R_{m1}, R_{m2}$; transform functions $f_{s1}, f_{s2}, f_{m1}, f_{m2}$
2: **while** not converged **do**
3:   **for** minibatch $(\mathbf{x}_1, \mathbf{x}_2)$ in $\mathcal{D}_{\text{train}}$ **do**
4:     $(\mathbf{h}_1, \mathbf{h}_2) \leftarrow E_1(\mathbf{x}_1), E_2(\mathbf{x}_2)$
5:     $\Phi_1 \leftarrow \mathbf{h}_1 + R_s^\top \big(f_{s1}(\mathbf{h}_1) - R_s\mathbf{h}_1\big)$
        $+ R_{m1}^\top \big(f_{m1}(\mathbf{h}_1) - R_{m1}\mathbf{h}_1\big)$
6:     $\Phi_2 \leftarrow \mathbf{h}_2 + R_s^\top \big(f_{s2}(\mathbf{h}_2) - R_s\mathbf{h}_2\big)$
        $+ R_{m2}^\top \big(f_{m2}(\mathbf{h}_2) - R_{m2}\mathbf{h}_2\big)$
7:     $\mathbf{z}_{s1} \leftarrow R_s\Phi_1, \quad \mathbf{z}_{m1} \leftarrow R_{m1}\Phi_1$
8:     $\mathbf{z}_{s2} \leftarrow R_s\Phi_2, \quad \mathbf{z}_{m2} \leftarrow R_{m2}\Phi_2$
9:     $\mathcal{L} \leftarrow \text{GradNorm}\big([L_{\text{orth}}, L_{\text{ind}}, L_{\text{mi}}]\big)$
10:     **Train:** $R_s, f_{s1}, f_{s2}, R_{m1}, R_{m2}, f_{m1}, f_{m2}$
11:   **end for**
12:   Evaluate validation losses $\mathcal{L}_{\text{val}}$ on $\mathcal{D}_{\text{val}}$
13:   **if** epoch $> 50$ **and** validation MI is within $1\%$ of the best validation MI **then**
14:     Adaptive rank pruning with threshold $\epsilon$ (Algo. 1)
15:   **end if**
16: **end while**

---

## 4. Experiments

Our experiments assess how effectively MultiLoReFT decouples information shared across modalities from modality-specific factors. We also show that the resulting fine-tuned representations serve as strong features for downstream tasks, capturing the joint signal for multimodal reasoning.

### 4.1. Datasets and Baselines

We first conduct controlled evaluations on simulated datasets, where the ground-truth generative structure is known. Each dataset includes conditional, joint, and unique labels, enabling targeted validation of different aspects of multimodal representation learning. We then scale to large, real-world datasets to assess the applicability of our method with pretrained encoders.

- Simulation I & II. Two synthetic multimodal datasets with controlled generative processes. Simulation I is constructed from independent latent factors drawn from diverse distributions, while Simulation II introduces dependencies across some factors, creating correlated shared and unique components. Both datasets provide labels that are modality-specific, shared, and conditional. Full details of dataset generation are presented in Appendix A.1.

- Flickr30K-Multi (Elliott et al., 2016). A multilingual extension of the Flickr30K containing image–caption pairs in five languages. Each image is paired with one caption per language, making language a modality-specific factor while semantic content remains shared. We use English and French captions for our experiments. For this dataset, we only use 1k out of 30k samples to assess few shot performance in a limited data setting.

- Crema-D (Cao et al., 2014). An audio–visual dataset of multimodal emotion expression and perception, comprising 7,442 clips from 91 actors across diverse demographics. Each clip contains both facial and vocal expressions of fixed sentences. The dataset provides both shared emotional signals and modality-specific cues.

- UR-FUNNY (Hasan et al., 2019). A multimodal humor detection dataset consisting of short video clips with aligned audio, visual, and textual modalities. We evaluate on two settings: (i) the commonly used benchmark version with pre-extracted modality representations, and (ii) a raw-video subset of 1933 clips, where we process visual streams directly from pretrained video models.

We use established pretrained models for each modality: DINO Vision Transformer for images (Caron et al., 2021), BERT-base for English text (Devlin et al., 2019), LaBSE for multilingual text (Feng et al., 2022), Wav2Vec 2.0 base pretrained on Librispeech-960h for audio (Baevski et al., 2020), and VideoMAE base model fine-tuned on Kinetics-400 for video embeddings (Tong et al., 2022). We compare our method against a broad set of multimodal representation learning frameworks, grouped into two categories:

- General fusion approaches. These methods integrate multimodal signals through concatenation, attention, or interaction mechanisms. As baselines, we use **late fusion** (Baltrušaitis et al., 2019), which simply concatenates modality representations, and an **attention**-based fusion model, which projects each modality into a common space and applies a lightweight self-attention layer to let the two embeddings interact before pooling into a fused representation. We also evaluate multiplicative interactions (**MI**) (Jayakumar et al., 2020), which extend tensor product fusion with learnable parameters to capture higher-order dependencies, and **contrastive learning**, which encourage aligned representations by pulling paired modalities closer and unpaired samples further.

- Decoupling approaches. These methods explicitly separate shared and modality-specific information. We consider **APOLLO** (Zhang et al., 2026), an autoencoder-based model that decouples shared and unique components through latent optimization, directly learning embeddings for training samples before training encoders

to generalize. We also benchmark DRIM (Robinet et al., 2024), which disentangles multimodal representations using three complementary objectives: enforcing similarity across shared embeddings, ensuring reconstruction fidelity, and adversarially regularizing unique modality-specific components. We adopt the self-supervised variant called **DRIM-U**, presented in the original paper, making it more comparable to our setting.

To ensure fairness and remove the effect of pretrained encoder selection, all baselines are trained on the same pretrained unimodal embeddings as input. The only architectural differences lie in the method-specific adapters, for example, DRIM-U uses a discriminator-based decoupling module, whereas APOLLO learns latent parameters directly.

## 5. Results

Our evaluation comprises three components: a representation analysis that quantifies how well shared and modality-specific information are decoupled, downstream multimodal prediction tasks that test multimodal performance, and multimodal performance under missingness to show cross-modal learning in fine-tuned representations.

### 5.1. Information Decoupling

The goal of disentangling shared and modality-specific signals is to assess the unique and common information that each modality contributes in a downstream multimodal setting. We compare MultiLoReFT against benchmark methods designed for disentanglement, and evaluate decoupling by measuring how each component predicts labels tied either to modality-specific or shared generative factors. Note that the representation learning is done fully unsupervised, and we only use the labels to train the regressor in order to assess the decoupling of information.

On the simulated datasets, we have ground-truth labels for the underlying generative factors. We train a logistic regressor on each representation component ($\mathbf{z}_s$, $\mathbf{z}_{m1}$, $\mathbf{z}_{m2}$) to predict these labels. This evaluation shows whether each factor is captured by its corresponding component and, how effectively it is removed from the remaining components.

Table 1 reports the decoupling results on simulated data, measured as the accuracy of a logistic regressor trained on each representation component. Underlined entries indicate the component that should best encode the underlying factor, and the $\Delta$ row reports the performance gap between the true component and the other components; larger gaps indicate stronger decoupling. All methods achieve highest accuracy by the appropriate component, suggesting that each can capture the relevant signal for a given factor. However, baselines often exhibit smaller $\Delta$ values, indicating the same factor remains predictive from other components as well

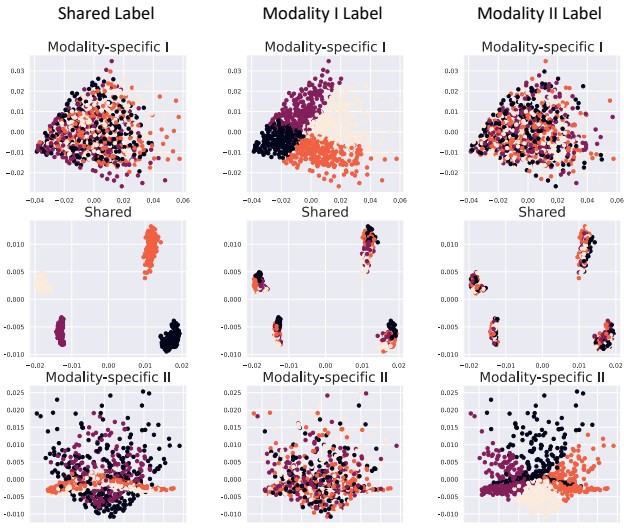

*Figure 2.* Visualization of subspaces learned by MultiLoReFT on Simulation I. Each panel shows a 2D PCA projection of the shared or modality-specific representations, colored by the underlying generative label. For the shared label (left columns), clear clustering emerges in the shared subspace while modality-specific subspaces remain unstructured. For the modality I and II labels, the corresponding modality-specific subspace captures the structure, whereas the shared subspaces remain agnostic.

(information leakage). This issue is most pronounced for the shared factor where MultiLoReFT yields the largest gap, while the baselines have similar performance on modality-specific and shared components. Decoupling shared information is particularly challenging because models often converge to a degenerate solution that encodes all information into modality-specific representations, which fails to learn and isolate the shared information. MultiLoReFT also achieves the best (or close second-best) separation for modality-specific factors. Finally, the baselines degrade significantly on Simulation II, where higher dimensionality and correlations among generative factors make disentanglement more difficult. Figure 2 visualizes the subspaces learned by MultiLoReFT, demonstrating that the shared label is clearly separable in the shared space, while each modality-specific label is best separated in its own subspace.

The results on real-world datasets (Table 2) further validate MultiLoReFT and highlight its utility on complex multimodal data. On Flickr30k-Multi, where the label indicates whether a caption is in English or French, the text-specific representation is the component that most reliably encodes language information. This behavior is also reflected in the well-separated language clusters in Figure 3(a). MultiLoReFT not only concentrates the language information in the correct subspace, but also achieves the largest $\Delta$ among all methods, indicating that it most effectively prevents this information from leaking into the shared space. In Fig-

*Table 1.* Decoupling of shared and modality-specific information on simulated data. Logistic-regression accuracy is reported for each representation component when predicting ground-truth generative factors. Larger $\Delta$ indicates stronger separation across components.

| | | Simulation I | | | Simulation II | |
|---|---|---|---|---|---|---|
| Model | Rep. | Shared | M1 | M2 | Shared | M1 |
| MultiLoReFT | $\mathbf{z}_s$ | 100.0±0.0 | 37.8±1.5 | 44.9±5.2 | 100.0±0.0 | 53.1±3.2 |
| | $\mathbf{z}_{m1}$ | 82.0±2.1 | 95.2±1.4 | 25.5±0.1 | 52.8±1.9 | 100.0±0.0 |
| | $\mathbf{z}_{m2}$ | 61.5±6.5 | 26.0±0.7 | 89.8±4.8 | 61.7±15.1 | 50.3±0.1 |
| | $\Delta$ | **18.0** | **57.4** | 44.9 | **38.3** | **46.9** |
| DRIM-U | $\mathbf{z}_s$ | 100.0±0.0 | 57.0±4.8 | 50.8±14.6 | 100.0±0.0 | 100.0±0.0 |
| | $\mathbf{z}_{m1}$ | 99.1±1.2 | 91.4±8.6 | 23.9±1.2 | 100.0±0.0 | 100.0±0.0 |
| | $\mathbf{z}_{m2}$ | 99.9±0.2 | 25.6±0.4 | 97.8±0.7 | 100.0±0.0 | 50.1±0.5 |
| | $\Delta$ | 0.1 | 34.4 | **47.0** | 0.00 | 0.00 |
| APOLLO | $\mathbf{z}_s$ | 100.0±0.0 | 75.1±1.0 | 71.8±9.9 | 100.0±0.0 | 100.0±0.0 |
| | $\mathbf{z}_{m1}$ | 100.0±0.0 | 93.0±1.3 | 26.3±0.4 | 100.0±0.0 | 100.0±0.0 |
| | $\mathbf{z}_{m2}$ | 100.0±0.0 | 24.8±0.4 | 97.4±1.5 | 100.0±0.0 | 50.9±0.1 |
| | $\Delta$ | 0.00 | 17.9 | 25.6 | 0.00 | 0.00 |

*Table 2.* Decoupling of shared and modality-specific information on real-world datasets (Crema-D and Flickr30). Performance is reported for each representation component ($\mathbf{z}_s, \mathbf{z}_{m1}, \mathbf{z}_{m2}$) on downstream labels corresponding to shared and modality-specific factors.

| | | CremaD | | | | | Flickr30 | |
|---|---|---|---|---|---|---|---|---|
| Model | Rep. | Sentence ID (Acc.)↑ | Race (Acc.) ↑ | Age (MSE)↓ | Sex (Acc.) ↑ | | Rep. | Language (Acc.)↑ |
| MultiLoReFT | $\mathbf{z}_s$ (Shared) | 68.5±8.1 | 50.1±1.5 | 131.7±12.4 | 90.1±2.3 | | $\mathbf{z}_s$ (Shared) | 77.2±7.3 |
| | $\mathbf{z}_{m1}$ (Video) | 24.5±0.1 | 85.7±13.2 | 55.8±8.8 | 69.6±2.5 | | $\mathbf{z}_{m1}$ (Image) | 50.5±0.2 |
| | $\mathbf{z}_{m2}$ (Audio) | 99.2±0.1 | 50.6±0.6 | 154.3±1.9 | 66.9±3.3 | | $\mathbf{z}_{m2}$ (Text) | 99.7±0.2 |
| | $\Delta$ | **30.7** | **35.6** | **75.9** | **20.5** | | | **22.5** |
| DRIM-U | $\mathbf{z}_s$ (Shared) | 74.7±17.0 | 75.2±2.4 | 118.7±27.4 | 94.8±1.6 | | $\mathbf{z}_s$ (Shared) | 79.6±9.8 |
| | $\mathbf{z}_{m1}$ Video | 19.9±1.2 | 69.9±3.5 | 118.3±12.9 | 84.9±0.2 | | $\mathbf{z}_{m1}$ (Image) | 48.9±02.0 |
| | $\mathbf{z}_{m2}$ (Audio) | 92.1±4.1 | 53.0±0.0 | 174.0±0.9 | 51.1±0.0 | | $\mathbf{z}_{m1}$ (Text) | 100.0±0.0 |
| | $\Delta$ | 17.4 | -5.3 | 0.4 | 9.9 | | | 20.4 |
| APOLLO | $\mathbf{z}_s$ (Shared) | 78.1±15.9 | 67.5±8.5 | 136.8±8.3 | 80.8±6.5 | | $\mathbf{z}_s$ (Shared) | 100.0±0.0 |
| | $\mathbf{z}_{m1}$ (Video) | 21.6±2.1 | 66.3±7.9 | 145.3±11.9 | 75.4±5.3 | | $\mathbf{z}_{m1}$ (Image) | 49.2±01.4 |
| | $\mathbf{z}_{m2}$ (Audio) | 78.2±16.4 | 53.5±0.5 | 176.2±6.0 | 70.7±6.3 | | $\mathbf{z}_{m1}$ (Text) | 100.0±0.0 |
| | $\Delta$ | 0.1 | -1.2 | -8.5 | 5.4 | | | 0.0 |

ure 3(b), a reference image is shown together with its nearest neighbors retrieved from the shared and image-specific subspaces. Neighbors in the shared space depict very similar high-level content: people cutting or working with materials in a workshop-like setting, closely matching the caption that emphasizes cutting wood. In contrast, neighbors in the image-specific space tend to cluster around visual details such as wood, logs, and trees, including outdoor scenes that are not mentioned in the description. This qualitative example illustrates how the shared subspace captures cross-modal semantic content, while the image-specific subspace emphasizes modality-specific visual cues.

On Crema-D (M1: video, M2: audio), we leverage the metadata to validate decoupling. We consider Sentence ID

(most reliably conveyed by the audio track), Age and Race (primarily encoded in the visual stream), and Sex (which can be inferred from both audio and video). Table 2 shows that MultiLoReFT consistently learns each attribute in the expected subspace, while yielding larger gaps between relevant and irrelevant components, indicating reduced leakage across factors. The scatter plots in Figure 3(a) further illustrate how information about different labels is spread within each subspace, with clearer separation along the dimensions corresponding to the appropriate component.

A key strength of MultiLoReFT is its adaptive rank learning, which automatically selects the effective dimensionality of each subspace and reduces the need for manual tuning. Subspaces are initialized at the unimodal representation

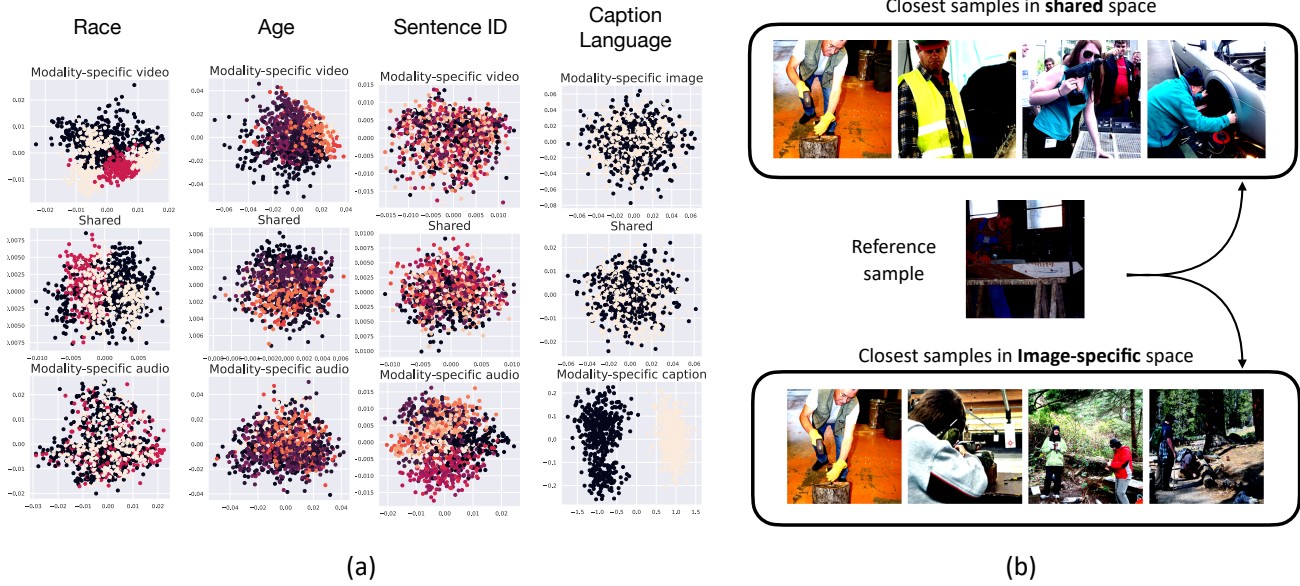

*Figure 3.* (a) PCA visualizations of MultiLoReFT representations on Crema-D and Flickr30. Task-relevant information is concentrated in the corresponding modality-specific components, while other components remain closer to random structure, illustrating effective disentanglement. (b) An example image from Flickt30 dataset with its closest samples in the image-specific subspace and shared subspace.

dimension, and during training, less-informative directions are progressively pruned, yielding lower-rank projections with concentrated information content. Appendix Table 5 reports the initial and converged ranks across experiments and Appendix A.4 further analyzes the resulting parameter efficiency of MultiLoReFT relative to existing baselines.

### 5.2. Multimodal Performance

MultiLoReFT is designed to decouple shared and modality-specific information, but it also yields representations that remain effective for multimodal prediction. The fine-tuned representations $\Phi$ preserve complementary signals that can be combined with simple fusion. As shown in Table 3, concatenating fine-tuned representations $\Phi_1$ and $\Phi_2$ and training a lightweight logistic regressor consistently outperforms a range of baselines, including disentanglement-based methods and classical fusion strategies such as late fusion, mutual-information–based objectives, and cross-attention. We also show that simple concatenation of fine-tuned MultiLoReFT representations achieves better or very similar performance to state-of-the-art (SOTA) models designed and trained for mutlimodal fusion. Finally, in Appendix A.5 we analyze the impact of the pretrained encoder quality on multimodal performance, showing that more expressive encoders yield systematically better multimodal representations. This highlights the flexibility of MultiLoReFT which can directly leverage improvements in unimodal pretraining to adapt and enhance multimodal models as more powerful encoders become available.

### 5.3. Missing Modality Performance

A key advantage of multimodal learning, particularly when the shared information across modalities is modeled explicitly, is improved robustness to missing modalities at inference time. In our framework, the fine-tuned representation of each modality is encouraged to capture not only modality-specific structure but also the information that is predictive of the other modality through the learned shared subspace. Concretely, each fine-tuned representation $\Phi$ can be projected into this shared space, and the training objective enforces it to preserve the cross-modal information. As a result, when one modality is missing, the fine-tuned representation of the remaining modality provides a stronger prediction signal than the corresponding pretrained embedding $h$. Figure 4 illustrates this effect across three datasets (Simulation I, Crema-D, and UR-Funny) for the multimodal prediction task. By comparing $h$ and $\Phi$ under two settings where one modality is missing, we show that the fine-tuned representations consistently improve downstream accuracy when either modality is absent.

## 6. Discussion

This work introduces MultiLoReFT, a low-rank representation fine-tuning framework for multimodal learning that explicitly disentangles shared and modality-specific information. The approach is model-agnostic, parameter-efficient, and operates on top of frozen pretrained unimodal encoders, enabling effective multimodal adaptation without sacrificing interpretability. MultiLoReFT also has an adaptive rank

*Table 3.* Multimodal prediction performance across multiple datasets. Results compare fusion of MultiLoReFT representations Φ against existing disentanglement-based and classical fusion baselines built on pretrained unimodal embeddings.

| Method | Simulation I | Crema-D | UR-Funny | UR-Funny (raw video) |
|---|---|---|---|---|
| Label | joint label | emotion | humor | humor |
| MultiLoReFT | **40.3±3.1** | **76.0±0.4** | **61.1±0.2** | **58.3±0.6** |
| APOLLO | 39.0±1.2 | 46.8±10.3 | 50.6±2.0 | 57.1±0.3 |
| DRIM-U | 38.7±1.2 | 69.3±0.8 | 59.3±0.4 | 54.6±1.5 |
| Late fusion | 32.7±0.0 | 74.9±0.0 | 58.8±0.0 | 57.6±0.0 |
| Cross attention | 33.0±0.4 | 72.6±1.6 | 60.7±0.2 | **58.2±0.9** |
| Contrastive | 26.3±3.6 | 69.9±0.0 | 49.7±0.0 | 54.9±1.0 |
| MI | 35.1±0.3 | 75.8±0.6 | 58.9±0.3 | 57.1±1.0 |
| SOTA models | — | MMPareto: 75.13 | FactorCL: 60.5 | — |
|  | — | (Wei & Hu, 2024) | (Liang et al., 2023b) | — |

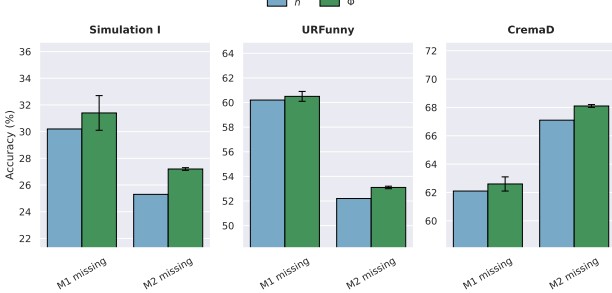

*Figure 4.* Downstream classification accuracy under missing-modality inference. For each dataset, we compare the pretrained representation $h$ and the fine-tuned representation Φ of the available modality when the other modality is missing. "M1 missing" evaluates prediction using only modality 2, and "M2 missing" evaluates prediction using only modality 1.

pruning mechanism which automatically selects the effective dimensionality of each subspace, improving both performance and insight into how information is distributed across modalities. The significance of MultiLoReFT lies in its ability to leverage strong unimodal pretrained models for multimodal downstream tasks. By separating shared structure from modality-specific cues, the framework provides interpretable insights into each modality's contribution while remaining practical in settings where paired multimodal data are scarce. These properties are particularly relevant in scientific and applied domains such as healthcare, where data collection is costly and interpretability is essential.

MultiLoReFT can be extended to more than two modalities by introducing additional projection subspaces and independence constraints. We demonstrate this on CMU-MOSI (Zadeh et al., 2016), a tri-modal dataset (video, audio, text) for sentiment prediction (Appendix A.6), showing that the learned representations yield improved performance over baselines on downstream multimodal tasks. However, such extensions raise interpretability challenges. In particular, it becomes unclear how to interpret partially shared factors (information shared between two modalities but not a third) or how such factors should be exploited in downstream tasks. Moreover, the lack of ground-truth labels for validating finer-grained decompositions complicates evaluation. For these reasons and for consistency with prior work, the present study is mainly restricted to bimodal data and future work can explore extension to more modalities and how to interpret various subspaces. Finally, because Multi-LoReFT builds on frozen pretrained unimodal encoders, its performance is ultimately bounded by the quality and coverage of those representations. When unimodal encoders fail to capture task-relevant factors, the benefits of fine-tuning are limited. Future work could explore integrating Multi-LoReFT with joint or continual pretraining, scaling to larger multimodal corpora, and developing principled methods to identify and overcome limitations of unimodal pretrained encoders during fine-tuning.

## Acknowledgements

S.T. and V.S. were supported by the Eric and Wendy Schmidt Center at the Broad Institute. V.S. was additionally supported by the Novo Nordisk Foundation grant NNF24OC0089461. C.U. was partially supported by NCCIH/NIH (1DP2AT012345), ONR (N00014-24-1-2687), and the United States Department of Energy (DE-SC0023187).

## Impact Statement

Multimodal models are especially valuable in high-stakes domains like healthcare, where decisions depend on combining heterogeneous data sources. Yet, truly paired multimodal datasets are difficult to collect in observational settings and are essential as the data is inherently long-tailed,

where rare conditions and underrepresented modality combinations dominate. This creates a practical barrier that strong multimodal systems often require far more data and parameters than real deployments can support. By providing an interpretable, parameter-efficient way to adapt strong unimodal models to multimodal tasks, this work lowers the data and compute cost of multimodal learning and enables more reliable, transparent models that can be validated for decision making.

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

# A. Appendix

## A.1. Simulated datasets

To systematically evaluate the functionality of our approach, we constructed two simulated datasets in which the underlying generative factors are explicitly controlled and well understood. These datasets enable us to verify whether each component of MultiLoReFT captures the intended signal. We provide a brief description of each below.

### A.1.1. SIMULATION I

Simulation I generates two modalities from a combination of shared and modality-specific latent variables, sampled from non-Gaussian distributions. In total, $n_{\text{hidden}} = 2 + 2 + 2 = 6$ hidden variables are defined: two shared, two private to modality 1, and two private to modality 2.

**Shared latent factors.** The shared variables $z_s \in \mathbb{R}^2$ are sampled from a *binomial* distribution and slightly perturbed with Gaussian noise to introduce variability:

$$z_s \sim \text{Binomial}(1, 0.5) + \mathcal{N}(0, 0.01I).$$

**Modality-specific latent factors.** Each modality has its own private factors drawn from distinct non-Gaussian distributions:

$$z_{m1} \sim \text{Weibull}(1.5) \times 0.3, \qquad z_{m2} \sim \text{Beta}(3, 2).$$

**Labels.** Each data point is annotated with four *categorical* labels to probe shared, modality-specific, and joint information:

$$y_{\text{shared}} \in \{0, 1, 2, 3\}, \quad y_{m1} \in \{0, 1, 2, 3\}, \quad y_{m2} \in \{0, 1, 2, 3\}, \quad y_{\text{joint}} \in \{0, 1, 2, 3, 4, 5, 6, 7\}$$

$y_{\text{shared}}$ is a multi-class label (0–3) determined by the unique binary configurations of the unnoised shared factors $z_s$. To obtain categorical modality-specific labels, the private continuous factors are first binarized dimension-wise using the empirical median threshold, yielding a 2-bit codeword per modality; $y_{m1}$ and $y_{m2}$ are then assigned as the index of the corresponding unique codeword (again producing up to four classes per modality). Finally, the joint label $y_{\text{joint}}$ is defined by the triple $(y_{\text{shared}}, y_{m1}, y_{m2})$: each unique triple is mapped to a joint class ID, and these IDs are deterministically bucketed (using a seed-controlled permutation) into at most $K = 8$ joint classes. This construction ensures $y_{\text{joint}}$ depends on *all* three other labels, and is not designed to be predictable from either modality alone.

### A.1.2. SIMULATION II

Simulation II generates two modalities from five latent variables with structured dependencies that introduce both shared and modality-specific components, as well as partial overlap.

**Shared latent factors.** Two variables define the shared information:

$$z_1 \sim \text{Ber}(0.5), \qquad z_2 = z_1 + \sqrt{0.0045}\,\Gamma(5, 1),$$

where $z_1$ is a binary Bernoulli variable and $z_2$ is a continuous variable correlated with $z_1$ through an additive Gamma perturbation.

**Modality-specific latent factors.** Modality 1 has two private factors:

$$z_3 \sim \text{Ber}(0.5), \qquad z_4 = 2z_2 + z_3 + \sqrt{0.00125}\,\Gamma(2, 2),$$

while Modality 2 has one private factor:

$$z_5 = z_2 + \sqrt{0.0075}\,\Gamma(3, 2).$$

This setup ensures overlap, as $z_2$ influences both $z_4$ and $z_5$, creating cross-modal dependencies while maintaining modality-specific variation.

**Labels.** Labels are directly tied to the latent variables, enabling controlled evaluation of shared versus modality-specific representations. Binary classification tasks can be derived from $z_1$ (shared) or $z_3$ (modality 1 specific), while regression targets can be defined from $z_2$ (shared), $z_4$ (modality 1 specific), or $z_5$ (modality 2 specific).

**Representations.** As in Simulation I, observed features for each modality are constructed by concatenating the corresponding shared and private variables and projecting them into higher-dimensional feature spaces via random linear transformations:

$$h_1 = [z_{m1}, z_s]W_1, \qquad h_2 = [z_{m2}, z_s]W_2,$$

where $W_1, W_2$ are sampled from uniform distributions. The output dimensionality is set to 40 and 80 for each modality.

This design complements Simulation I by introducing structured overlap and dependence between modalities, testing whether models can disentangle shared information from modality-specific signals in the presence of cross-modal dependencies.

### A.2. Ablation study

To assess the contribution of individual design choices in MultiLoReFT, we conduct ablation experiments removing design componenets like the pruning mechanism or loss weighting through GradNorm. Results are summarized in Table 4 showing both components are critical: eliminating pruning leads to inflated subspace sizes and information leakage across components, while GradNorm helps with a more stable convergence without requiring any hyper-parameter selection. The full model, consistently achieves the highest decoupling, underscoring their complementary benefits.

Furthermore, Table 4 shows the effect of different components of the loss term in the overall performance of MultiLoReFT. Each row presents the results with one component removed, and we see that the full model achieves the highest consistent performance. Mutual-information (MI) retention loss preserves unimodal content and without it (MultiLoReFT - MI), overall performance drops across all subspaces (shared and private).

For the decoupling, the orthogonality loss alone (MultiLoReFT - independence) seems is sufficient for modality-specific signals, since these targets depend mainly on unimodal geometry. Here, each modality's signal lies on its own manifold, so ensuring linear disjointness prevents interference without needing additional statistical constraints. However, the shared categorical label relies on both orthogonality and independence for successful decoupling, because information emerges from the joint statistical structure across modalities; orthogonality separates the spaces geometrically, while independence removes nonlinear correlations and redundancy, allowing the shared subspace to capture only the truly cross-modal information rather than correlated modality-specific noise.

*Table 4.* Ablation of the MultiLoReFT objective. The first row is the full model. The next three rows remove the indicated loss term (one at a time). The final row trains with fixed loss weights instead of automatic weighting via GradNorm.

| | | Simulation I | | | Simulation II | |
|---|---|---|---|---|---|---|
| Model | Rep. | Shared (Acc.)↑ | M1 (Acc.)↑ | M2 (Acc.)↑ | Shared (Acc.)↑ | M1 (Acc.)↑ |
| MultiLoReFT | $\mathbf{z}_s$ | 100.0±0.0 | 37.8±1.5 | 44.9±5.2 | 100.0±0.0 | 53.1±3.2 |
| | $\mathbf{z}_{m1}$ | 82.0±2.1 | 95.2±1.4 | 25.5±0.1 | 52.8±1.9 | 100.0±0.0 |
| | $\mathbf{z}_{m2}$ | 61.5±6.5 | 26.0±0.7 | 89.8±4.8 | 61.7±15.1 | 50.3±0.1 |
| MultiLoReFT | $\mathbf{z}_s$ | 100.0±0.0 | 57.5±8.0 | 63.8±7.2 | 100.0±0.0 | 91.6±11.8 |
| - pruning | $\mathbf{z}_{m1}$ | 86.5±16.7 | 96.0±0.7 | 23.3±0.3 | 68.1±1.0 | 82.7±12.2 |
| | $\mathbf{z}_{m2}$ | 90.4±7.6 | 24.0±1.9 | 86.3±0.8 | 79.6±18.9 | 50.8±0.6 |
| MultiLoReFT | $\mathbf{z}_s$ | 100.0±0.0 | 82.2±06.0 | 76.3±9.6 | 100.0±0.0 | 75.3±2.2 |
| - GradNorm | $\mathbf{z}_{m1}$ | 100.0±0.0 | 95.6±1.7 | 24.4±1.4 | 63.8±2.3 | 99.7±0.4 |
| | $\mathbf{z}_{m2}$ | 100.0±0.0 | 25.1±0.9 | 92.9±4.7 | 93.5±5.0 | 51.0±0.1 |
| MultiLoReFT | $\mathbf{z}_s$ | 99.9±0.1 | 68.7±19.4 | 44.9±11.8 | 99.5±0.6 | 79.0±17.3 |
| - MI | $\mathbf{z}_{m1}$ | 91.0±6.2 | 67.2±1.9 | 24.0±0.4 | 78.8±21.3 | 100.0±0.0 |
| | $\mathbf{z}_{m2}$ | 64.6±18.4 | 24.3±1.7 | 92.3±1.6 | 98.8±1.7 | 50.9±0.1 |
| MultiLoReFT | $\mathbf{z}_s$ | 100.0±0.0 | 34.3±4.2 | 40.2±5.3 | 100.0±0.0 | 0.71.9±0.15.6 |
| - Independence | $\mathbf{z}_{m1}$ | 100.0±0.0 | 96.6±0.5 | 23.9±0.2 | 100.0±0.0 | 100.0±0.0 |
| | $\mathbf{z}_{m2}$ | 100.0±0.0 | 25.1±0.6 | 94.1±2.0 | 98.3±2.3 | 50.6±0.1 |
| MultiLoReFT | $\mathbf{z}_s$ | 99.9±0.2 | 72.6±5.9 | 72.5±12.3 | 94.8±7.4 | 58.7±3.0 |
| - Orthogonality | $\mathbf{z}_{m1}$ | 99.9±0.1 | 96.6±0.8 | 24.0±0.5 | 82.0±12.9 | 100.0±0.0 |
| | $\mathbf{z}_{m2}$ | 100.0±0.0 | 25.6±0.5 | 94.5±1.2 | 74.2±19.5 | 51.1±0.1 |

## A.3. Rank-adaptation

Table 5 reports the initial and converged dimensionalities of the shared ($\mathbf{z}_s$) and modality-specific ($\mathbf{z}_{m1}, \mathbf{z}_{m2}$) subspaces across different datasets. These results are averaged over multiple random seeds, with standard deviations shown to reflect variability. We observe that MultiLoReFT consistently prunes high-dimensional initializations down to compact and stable subspaces, with only minor variation across runs. This consistency highlights that the model is able to reliably identify the rank of shared versus modality-specific information.

The results presented in the main paper are achieved with these compact representations, demonstrating that strong disentanglement and predictive performance do not require large subspace sizes. Instead, the rank adaptation procedure ensures both efficiency and interpretability, by automatically converging to low-dimensional but informative representations across datasets.

*Table 5.* Shared and modality-specific subspace dimensionalities learned by MultiLoReFT. Entries show the initial rank $\rightarrow$ converged rank mean and standard deviation on 4 different random seeds

|  | Simulation I | Simulation II | Crema-D | Flickr |
|---|---|---|---|---|
| $\mathbf{z}_s$ | $10 \rightarrow 4.6 \pm 0.47$ | $40 \rightarrow 9.3 \pm 3.09$ | $700 \rightarrow 32.0 \pm 1.41$ | $700 \rightarrow 6.6 \pm 3.1$ |
| $\mathbf{z}_{m1}$ | $10 \rightarrow 4.3 \pm 0.47$ | $40 \rightarrow 4.6 \pm 1.63$ | $700 \rightarrow 40.6 \pm 1.69$ | $700 \rightarrow 27.0 \pm 3.7$ |
| $\mathbf{z}_{m2}$ | $10 \rightarrow 5 \pm 0.81$ | $40 \rightarrow 5.0 \pm 1.63$ | $700 \rightarrow 42.3 \pm 2.05$ | $700 \rightarrow 21.6 \pm 3.7$ |

## A.4. Parameter size comparison

A central motivation behind PEFT methods is to achieve *parameter-efficient fine-tuning*. Rather than updating all weights of large pretrained encoders, recent methods introduce lightweight modules whose size scales with the input representation dimension $d$ and the learned subspace size $d^*$. This allows fair comparison across benchmarks in terms of their parameter overhead.

For **MultiLoReFT**, the number of trainable parameters is on the order of:

$$\#\text{params} \approx 770 \, d \, d^*,$$

corresponding to the projection matrices and small transformation functions. Importantly, we begin with a large parameter space but prune down subspaces dynamically during training, which further reduces the effective size.

For **APOLLO**, the parameter cost includes adaptor layers and explicit sample-wise representations, yielding:

$$\#\text{params} \approx 2048 \, d \, d^* \; + \; 3d^* n_{\text{train}},$$

where the second term scales linearly with the number of training samples $n_{\text{train}}$, making the method less efficient for large datasets.

For **DRIM-U**, adaptor heads, decoders, and discriminators introduce higher overhead:

$$\#\text{params} \approx 1536 \, d \, d^* \; + \; 65.5d^*.$$

These expressions approximate $d^*$ as the average subspace size across shared and modality-specific components. While exact sizes may vary, the relative scaling highlights the efficiency of MultiLoReFT, enabling scalable fine-tuning in multimodal settings and making training feasible under limited computational budgets.

## A.5. Effect of Pretrained Unimodal Encoder Strength on Multimodal Performance

Table 6 compares multimodal performance on different pre-trained encoders on the CREMA-D dataset, emotion recognition task. In the first configuration **(Encoders I)**, we use simpler unimodal encoders, Wav2Vec 2.0 Base pretrained on Librispeech-960h (Baevski et al., 2020) for audio and 3D ResNet-18 pretrained on Kinetics-400 (He et al., 2016; Kay et al., 2017) for video. In the second configuration **(Encoders II)**, we replace these with stronger encoders, MViT-V2-S pretrained on Kinetics-400 (Li et al., 2022) for video and WavLM-Base+ for audio (Chen et al., 2022).

As shown, MultiLoReFT performance improve with higher-capacity unimodal encoders, which is an advantage of methods that leverage pretrained encoders. As unimodal models continue to advance, their improved representational quality can be benefited from to construct higher-performing multimodal representations, even under limited multimodal supervision.

*Table 6.* The effect of Unimodal encoder strength on multimodal Performance on Crema-D dataset for emotion detection. Encoders I setup uses relatively simpler video and audio encoders while Encoders II setup uses more advanced pretrained models.

| Method | Crema-D emotion Encoders I | Crema-D emotion Encoders II |
|---|---|---|
| MultiLoReFT | 46.0±3.1 | 76.0±0.4 |
| APOLLO | 31.3±1.2 | 46.8±10.3 |
| DRIM-U | 44.3±1.8 | 69.3±0.8 |
| Late fusion | 45.2±3.9 | 74.9±0.0 |
| Cross attention | 30.4±3.2 | 72.6±1.6 |
| Contrastive | 41.9±2.5 | 69.9±0.0 |
| MI | 29.6±2.2 | 75.8±0.6 |

### A.6. MultiLoReFT and 3 modalitites

The core formulation of MultiLoReFT extends naturally to settings with more than two modalities. For $M$ modalities, the shared subspace is defined as the component of information common to all modalities, while each modality also retains a private subspace capturing modality-specific variation. In the main paper, we focus on the bimodal case because it permits a cleaner and more controlled evaluation of disentanglement. In particular, the distinction between shared and private factors is easier to analyze and validate.

To demonstrate that the framework also applies beyond the bimodal case, we evaluate MultiLoReFT on CMU-MOSI (Zadeh et al., 2016), which contains three modalities (video, audio, and text) for sentiment prediction. Since this dataset does not provide ground-truth labels for shared and modality-specific factors, we do not evaluate disentanglement directly. Instead, we assess the quality of the fine-tuned representations through downstream multimodal prediction performance. For video and audio, we use the released pretrained features, since the raw videos and audio recordings are not available. For text, we use the raw data with a BERT-base embedding model (Devlin et al., 2019) (similar to Flickr experiments). We show that the fine-tuned representation improve binary classification accuracy for sentiment analysis by 2 percent ($54.81 \pm 0.00$ to $56.84 \pm 00.04$) compared to the pretrained embeddings.

If finer-grained explainability is desired, the framework could be extended further to include additional subspaces corresponding to information shared by subsets of modalities, such as pairwise-shared components. In that case, the orthogonality and independence constraints would also need to be generalized to account for these intermediate shared subspaces.

