# OpenReview forum: "MultiLoReFT: Decoupling Shared and Modality-Specific Subspaces in Multimodal Learning via Low-Rank Representation Fine-Tuning"
_ICML.cc/2026/Conference — ICML 2026 regular_

### Official Review · Reviewer_vehk · 2026-03-12

**Soundness:** 1
**Presentation:** 3
**Significance:** 4
**Originality:** 2
**Overall Recommendation:** 3
**Confidence:** 3

**Summary:**

The paper addresses key challenges in multimodal learning, where data from different modalities—such as images, text, and audio—must be processed jointly. Specifically, it targets two major issues: data scarcity and entangled representations.
To tackle these challenges, the authors propose MultiLoReFT, a low-rank representation fine-tuning approach. This method efficiently leverages pre-trained single-modality models, decouples shared information from modality-specific information, and maintains both parameter efficiency and model interpretability.
In essence, the paper focuses on adapting pre-trained single-modality models to multimodal tasks in a way that preserves interpretability and parameter efficiency while explicitly separating shared and modality-specific representations.

**Compliance With Llm Reviewing Policy:**

Affirmed.

**Key Questions For Authors:**

the disentanglement experiments have limited evidential strength. Synthetic data is constructed around shared and private latent factors, naturally favoring explicit subspace decomposition. Real-data validation also relies on strong prior assumptions—for example, in Crema-D, Sentence ID is treated as audio-dominant, Age and Race as visual-dominant, and Sex as shared. While reasonable, these assignments are not rigorously justified, so Table 2 provides evidence consistent with the authors’ assumptions rather than an unambiguous demonstration of disentanglement.

**Limitations:**

yes

**Strengths And Weaknesses:**

Soundness
The authors build on frozen pre-trained encoders and explicitly learn three low-rank subspaces: one shared subspace and two modality-specific subspaces. These are constrained by three losses—independence (HSIC), orthogonality, and cross-modal mutual information preservation. Training further incorporates GradNorm for multi-objective balancing and SVD-based adaptive rank pruning. Overall, the design logic is clear and coherent.
However, from a rigor perspective, the paper has several key limitations. First, the core claims are not fully supported by the experiments. The introduction asserts that the method “improves generalization across domains” and “better handles missing modalities,” yet the experiments do not directly test cross-domain generalization or robustness to missing modalities. The evaluation is limited to decoupling probes and standard downstream tasks, which is insufficient to substantiate these broader claims.
Second, the disentanglement experiments have limited evidential strength. Synthetic data is constructed around shared and private latent factors, naturally favoring explicit subspace decomposition. Real-data validation also relies on strong prior assumptions—for example, in Crema-D, Sentence ID is treated as audio-dominant, Age and Race as visual-dominant, and Sex as shared. While reasonable, these assignments are not rigorously justified, so Table 2 provides evidence consistent with the authors’ assumptions rather than an unambiguous demonstration of disentanglement.
Third, the method’s complexity may outweigh its empirical gains. It introduces multiple losses, GradNorm, warm-up, validation-MI-based pruning heuristics, and stepwise rank pruning, yet downstream performance improvements are modest. On Crema-D, MultiLoReFT only slightly exceeds late fusion (73.4 vs. 73.1), and on UR-FUNNY (raw video) it marginally outperforms cross-attention (58.3 vs. 58.2). This raises questions about whether the additional training mechanisms yield substantial benefit.
Finally, while the appendix includes ablations and claims that “all components are necessary,” the main text does not transparently quantify the independent contributions of each loss, GradNorm, or pruning heuristic. Practical choices such as 50-epoch warm-up, pruning triggered within 1% of validation MI, and a 10% maximum prune per step are clearly engineering-driven, yet the robustness of these hyperparameters is not sufficiently discussed.

Presentation
The paper is generally well-organized. The introduction clearly presents the problem context, including multimodal data scarcity, entanglement of shared and modality-specific information, and the high cost of full fine-tuning. The related work is sensibly divided into three areas: multimodal representation learning, disentangled multimodal representations, and multimodal fine-tuning. The methodology section is structured logically, and Figure 1 provides an intuitive illustration of the shared and private subspace workflow, helping readers quickly understand how low-rank representations are used to achieve multimodal disentanglement.
However, the paper does not fully clarify its fine-grained distinctions from closely related work. While it cites DRIM-U, APOLLO, FactorCL, Triple Disentanglement, and LoReFT, the discussion of how the current approach differs from these methods remains high-level. As a result, although the paper is readable, its boundaries of innovation are not clearly communicated. Readers may perceive the contribution primarily as a structured combination of low-rank PEFT, multimodal disentanglement, and rank pruning, rather than as a novel, distinct perspective.
Additionally, the paper repeatedly emphasizes capabilities such as interpretability, handling missing modalities, and cross-domain generalization. The experiments, however, do not systematically support each of these claims, which risks overstating the work’s contributions and can affect the credibility of the presentation.

Significance
The problem studied in this paper is important. Effectively leveraging strong single-modality pre-trained models to build efficient and interpretable multimodal systems is a key challenge in current multimodal learning. This is particularly relevant in domains such as healthcare, where paired multimodal data are scarce and interpretability is critical. The discussion and impact statement appropriately situate the method within “data-scarce settings” and “high-stakes domains,” giving the work clear practical relevance. Even if performance gains are modest, the approach provides a useful design template for the intersection of multimodal PEFT and disentanglement, highlighting its potential value.
At the same time, the actual impact is limited. While the paper addresses a broad problem, the results primarily demonstrate feasibility rather than transformative performance. Downstream improvements are modest, offering limited evidence of practical benefit. The method is currently evaluated only on bimodal settings, and scaling to multiple modalities would raise challenges in handling partially shared factors and assessing interpretability. Its effectiveness is also constrained by the quality of the underlying single-modality encoders; if key signals are not captured, low-rank fine-tuning has limited potential. Overall, the contribution is best seen as a feasible exploration along a specific technical route rather than a substantial advancement for multimodal learning in practice.

Originality
The paper presents a moderately novel integration, applying low-rank representation fine-tuning to multimodal settings and combining it with shared/private subspace disentanglement, unsupervised structural constraints, and adaptive rank pruning. This originality primarily arises from the creative combination of existing ideas rather than the introduction of entirely new modules. Under a broad definition of originality, this contribution is reasonable.
However, the significance of this novelty is limited. Low-rank PEFT and LoReFT, shared/private disentanglement, and rank pruning are all established techniques. The core contribution of the paper is essentially combining these three approaches for multimodal representation learning with frozen single-modality encoders, implemented via HSIC, orthogonality, and mutual information constraints. While this is a thoughtful design, it does not clearly constitute a breakthrough for the field.
More importantly, the paper does not fully convince the reader that this combination represents a genuinely necessary or non-obvious modeling choice rather than a natural engineering integration. The differences from the most closely related work exist but are not sharply articulated, leaving the overall originality at a moderate rather than strong level.

---

> ### Author Rebuttal · Authors · 2026-03-30
>
> We appreciate your thoughtful feedback and thorough review, including the valuable comments and suggestions. We will ensure the manuscript is updated to reflect this input. Below, we address your concerns and comments:
>
>
> **Interpretability, generalization, and handling missingness:** Thank you for raising this concern. We will ensure these are clarified in the text. In summary, the interpretability claim is supported primarily through the decoupling experiments, since the central notion of interpretability in our setting is the ability to separate shared from modality-specific factors and to verify that these subspaces capture the intended information.
>
> Regarding generalization, our argument is that MultiLoReFT performs this factorization in an unsupervised way, without tying the decomposition to task labels. This is an important distinction from PID-based approaches that define shared information relative to downstream tasks. MultiMoReFT is therefore better positioned to transfer across various settings. That said, we agree that the current experiments do not directly validate cross-domain generalization benchmark.
>
> Regarding missingness, We would like to provide additional results that show how MultiLoReFT fine tuning helps with handling missing modalities.  Please see our response to Reviewer 4XEn for additional results on this.
>
> **Evaluating decoupling:** We would like to emphasize that controlled synthetic experiments are standard practice in disentanglement research precisely because they provide identifiable ground truth. We do not believe that having known shared and private factors “naturally favors” our method, especially since all other benchmarks are also evaluated on the data. This rather provides the only setting in which one can directly test whether a method recovers the intended structure.
>
> For the real-data experiments, we also agree that the underlying latent factors are not fully observed, so the evaluation cannot be interpreted as a definitive proof of disentanglement. However, the factors used in Table 2 are not arbitrary assumptions. They were chosen specifically in well-understood settings where there is a strong basis for treating them as modality-specific or shared proxies. For example, caption language is inherently text-specific and is not present in the image modality. We therefore view Table 2 not as evidence that the learned subspaces align with meaningful modality-relevant structure on real data.
>
> More broadly in real applications, the true generating factors are rarely known and the value of our method is also in enabling interpretability, explanation, and discovery in realistic settings where the latent structure is only partially known.
>
>
> **Multimodal performance gain:** We agree that the downstream gains are modest in some settings, however, at the same time, we emphasize that the main goal of MultiLoReFT is not simply to maximize fusion accuracy, but to learn more interpretable multimodal representations by separating shared and modality-specific information. On this axis, the method shows a clearer and more consistent advantage over the baselines. We view the downstream experiments as complementary evidence to show that introducing this additional structure does not come at the expense of representation quality. We will revise the text to make this framing clearer.
>
> **Ablation:** The appendix isolates the contribution of the major components through targeted ablations, and shows that the full model achieves the strongest overall performance. In particular, it is explained how removing individual components leads to consistent degradation. We will make sure that this is stated more clearly in the main text.
>
>
> **Related work and Novelty:** We are more than happy to clarify this in the text. At a high level, the key differences are two fold. First, methods such as Triple Disentanglement and FactorCL rely on supervised formulations, often grounded in PID-style objectives. They define the “shared” component relative to information jointly predictive of both modality and label. MultiLoReFT, like DRIM-U and APOLLO, aims to decouple representations in an unsupervised manner.
>
>  Second, among unsupervised methods, the decoupling mechanisms differ substantially. APOLLO relies on latent optimization, requiring learned latent variables for all samples, which makes scaling more challenging and DRIM-U enforces decoupling through an additional discriminator. MultiLoReFT instead performs decoupling through low-rank interventions on frozen pretrained representations. This makes the method lighter-weight, while still producing stronger decoupling empirically. We will revise the related-work to make the boundary of innovation more explicit.
>
>
> **Scalability beyond 2 modalities:** Thank you for this comment. Please see our response to Reviewer EQcL  for supplementary results.
>
> **Dependency on Pretrained Encoder Quality:** Please see our response toReviewer EQcL.

---

> > ### Author Rebuttal · Reviewer_vehk · 2026-04-02
> >
> > Thank you for the response to my review. I appreciate the clarification and the more careful framing of the claims. However, my main concern remains: the disentanglement experiments still have limited evidential strength, as the synthetic setting may favor explicit subspace decomposition and the real-data validation continues to rely on proxy assignments that are not rigorously justified. Therefore, I will keep my current score.

---

> > > ### Author Response · Authors · 2026-04-04
> > >
> > > We thank the reviewer for the follow-up, but we believe this criticism slightly mischaracterizes the evaluation issue. The absence of stronger real-data disentanglement evidence is not a limitation of our method; it is a fundamental limitation of the problem setting itself. In real data, the underlying generative factors are not observed, so there is no direct way to rigorously validate disentanglement without controlled synthetic factors or some form of proxy supervision. For this reason, simulation-based evaluation is the standard methodology in the disentanglement literature.
> > >
> > >
> > > Synthetic simulations are the only environment where ground-truth factors are known and can be validated. Verifying identifiability on synthetic data is a sanity check generally used in disentanglement papers [1,2]. Success here proves the method is logically consistent and provides merit by comparing directly with baseline and SOTA methods.
> > >
> > >
> > > On real-world data where the true generating factors are not known, the community has converged on using well-understood semantic attributes as proxies as a best practice [3,4]. Accordingly, our evaluation follows the established and appropriate protocol for this problem: we use simulation where ground-truth factors are available, and we use carefully motivated proxies on real data where they are not.
> > >
> > >
> > > [1] Eastwood, C. and Williams, C.K., 2018, February. A framework for the quantitative evaluation of disentangled representations. In the International conference on learning representations.
> > >
> > > [2]Daunhawer, I., Bizeul, A., Palumbo, E., Marx, A. and Vogt, J.E., Identifiability Results for Multimodal Contrastive Learning. In The Eleventh International Conference on Learning Representations.
> > >
> > > [3] Lee, M. and Pavlovic, V., 2021. Private-shared disentangled multimodal vae for learning of latent representations. In Proceedings of the ieee/cvf conference on computer vision and pattern recognition (pp. 1692-1700).
> > >
> > > [4] Peri, R., Parthasarathy, S., Bradshaw, C. and Sundaram, S., 2021, June. Disentanglement for audio-visual emotion recognition using multitask setup. In ICASSP 2021-2021 IEEE International Conference on Acoustics, Speech and Signal Processing (ICASSP) (pp. 6344-6348). IEEE.

---

### Official Review · Reviewer_4XEn · 2026-03-13

**Soundness:** 3
**Presentation:** 3
**Significance:** 3
**Originality:** 2
**Overall Recommendation:** 3
**Confidence:** 3

**Summary:**

The paper introduces MultiLoReFT, a parameter-efficient framework designed to fine-tune multimodal representations by learning decoupled subspaces for shared and modality-specific information using frozen unimodal encoders. The approach utilizes low-rank projection matrices to define these subspaces and applies localized transforms to refine the representations. The training process employs an unsupervised objective function that integrates HSIC-based independence, subspace orthogonality, and an InfoNCE-style mutual information term to ensure effective separation. An adaptive SVD-based rank pruning mechanism is also used to automatically determine the optimal dimensionality for each subspace. Experimental results across both simulated and real-world datasets indicate that MultiLoReFT achieves improved decoupling of shared vs. modality-specific factors and competitive performance compared to existing multimodal fusion and disentanglement baselines.

**Compliance With Llm Reviewing Policy:**

Affirmed.

**Key Questions For Authors:**

Please look at the weakness section.

**Limitations:**

yes

**Strengths And Weaknesses:**

Here are some strengths of the paper:

1. The formulation of representation-level, low-rank interventions to induce explicitly parameterized shared and private subspaces is a very good extension of LoRA ideas to multimodal disentanglement. In addition, finetuning already available pretrained unimodal representation models is a huge benefit for multimodal learning. The loss functions target the extraction of both modality-specific and shared representations.

2. The high-level motivation and architecture are well-motivated, and the training objective is clearly decomposed into intuitive terms. The depiction of subspaces and nearest-neighbor retrievals in shared vs. modality-specific spaces helps illustrate interpretability claims.

3. Addressing disentanglement of shared and private factors with a PEFT approach is relevant for advancing multimodal representation research, as many domains rely on frozen unimodal encoders and have limited paired data. The method’s focus on interpretable subspaces and the potential to handle missing modalities are practically important directions for multimodal systems.

4. The empirical results show good improvement and show that the proposed method is indeed very good.

Here are some weaknesses of the paper:

1. There are some works in the multimodal VAE space that address similar multimodal representation works but have not been addressed in this work or discussed in related works. Discussing those would be helpful in my opinion.

2. How does the model perform in missing modality scenarios at inference time? If one modality is missing, can the shared subspace be reliably reconstructed from the remaining modality to maintain prediction stability?

3. The current experiments are mostly bimodal. I believe it's good to have experiments with multiple modalities above 2.

---

> ### Author Rebuttal · Authors · 2026-03-30
>
> Thank you for your thoughtful and constructive feedback. We appreciate your recognition of the paper’s novel extension of low-rank adaptation to multimodal disentanglement, its practical value in leveraging pretrained unimodal encoders, the clarity of the motivation and training objectives, and the strong empirical results. We will use your comments to further strengthen the paper, and below we address your points in detail.
>
>
> **Multimodal VAE:** These works were not included in the current draft because our related work discussion was written to focus more on methods most directly connected to pre-trained unimodal models that avoid large training for components like a decoder. However, we agree that multimodal VAE-based approaches are relevant prior work and we will therefore add this line of work, including [1] and [2], to the related work section. We will also clarify the distinction from our setting: unlike multimodal VAE approaches, our method is centered on leveraging pretrained foundation-model embeddings and learning disentangled structure without introducing additional generative components such as decoders.
>
> [1] Lee M, Pavlovic V. Private-shared disentangled multimodal vae for learning of latent representations. InProceedings of the ieee/cvf conference on computer vision and pattern recognition 2021 (pp. 1692-1700).
>
> [2] Meo C, Lanillos P. Multimodal vae active inference controller. In2021 IEEE/RSJ International Conference on Intelligent Robots and Systems (IROS) 2021 Sep 27 (pp. 2693-2699). IEEE.
>
> **MultiLoReFT and missing modality:** Thank you for raising this important point. A benefit of learning a shared space is to learn to embed the cross-modal information. Therefore, at inference, even if a modality is missing, the shared information can be leveraged to improve prediction.
> We performed additional experiments to show how fine-tuned representations can be used when a modality is missing. The table below compares the predictive performance of the pre-trained and fine-tuned representations on a downstream prediction task, where one modality is missing. We observe in all cases that  fine-tuned representations perform better than pretrained embeddings. For Simulation II, shared label performances are generally high, because the label information is present in both modalities.In Simulation I however, joint labels show an overall low performance with missing modality, because the labels depend on information that exists in both modalities.  We will expand these results to other datasets as well and include this analysis in the results section.
>
> | |Shared H|Shared Phi|Joint H|Joint Phi|
> |---|---:|---:|---:|---:|
> |M2 missing|99.7±0.0|100.0±0.0|25.3±0.0|26.9±0.01|
> |M1 missing|99.9±0.0|100.0±0.0|30.2±0.0|31.4±0.0|
>
>
> **Scalability beyond 2 modalities:** MultiLoReFT is not fundamentally limited to two modalities and the same core formulation extends naturally to larger multimodal settings, as noted in the paper. The same principle applies in more than 2 modalities: the shared subspace is defined as the information shared across all modalities, while each modality also retains its own modality-specific subspace. The same losses extend accordingly: orthogonality and independence are imposed between the shared and modality-specific subspaces, and the mutual-information objective is used in the same way to preserve informative content while encouraging alignment of the shared representation across modalities.
>
> Our current study focuses on the bimodal setting because it allows a cleaner and more rigorous evaluation of disentanglement. To address the reviewer’s concern more directly, we will add an additional experiment on CMU-MOSI using all three modalities (video, audio, and text), for sentiment detection. Because MOSI does not provide ground-truth labels for shared and modality-specific factors, we will evaluate the quality of the learned representation through downstream multimodal prediction performance (Accuracy in binary classification), alongside the same baselines used elsewhere in the paper. For the rebuttal, we use the pretrained features released with the dataset, and simple logistic regression for the sentiment classification. For the rebuttal, we use the pretrained features released with the dataset together with a simple logistic-regression classifier for sentiment prediction. At this stage the resulting performance is therefore not directly comparable to task-specific SOTA models trained on raw inputs with stronger multimodal architectures. Our goal in this experiment is to show that a fully unsupervised MultiLoReFT factorization of the pretrained representations can still improve downstream sentiment prediction in datasets with more than 2 modalitties. In the revised version, we will extend this analysis to stronger pretrained foundation encoders and, where feasible, to raw-input settings as well.
>
> Late-fusion: 66.90 +- 0.00
> MultiLoReFT: 68.29 +- 0.00

---

> > ### Author Rebuttal · Reviewer_4XEn · 2026-04-03
> >
> > I thank the authors for the clarification. I still think a more robust experimental evaluation and discussion with MVAEs would benefit the paper. For that reason, I will keep my current score.

---

> > > ### Author Response · Authors · 2026-04-04
> > >
> > > Thank you for taking the time to review our rebuttal and we are very happy that the responses have clarified all your concerns.
> > >
> > >
> > > We would like to again address your remaining concern about MVAEs. We appreciate you bringing up that our related work did not explicitly cover this class of model. As stated in our rebuttal, we are expanding our related work section to include MVAEs in the discussion.
> > >
> > >
> > > We also would like to note that APOLLO, which is included in our benchmarks, is functionally very similar to a MVAE. It includes an encoder, decoder, and a distribution approximation for the decoupled latent representations. The only distinction is that APOLLO uses latent optimization instead of variational inference for training (which prior work shows performs similarly to, and in some cases better than variational inference in representation learning settings [1,2]). Specifically, the APOLLO paper motivates this choice by arguing that the variational objective is not sufficient to prevent a degenerate solution in which all information is pushed into the modality-specific space while the shared space contains only noise. We hope this helps clarify our choice of baselines, and that the reviewer may reconsider their score if their other concerns are satisfactorily addressed.
> > >
> > >
> > > [1] P. Bojanowski, A. Joulin, D. Lopez-Pas, and A. Szlam, “Optimizing the Latent Space of Generative Net works,” in Proceedings of the 35th International Conference on Machine Learning. PMLR, Jul. 2018, pp. 600–609.
> > >
> > >
> > > [2] Schuster, V. and Krogh, A., 2023. The Deep Generative Decoder: MAP estimation of representations improves modelling of single-cell RNA data. Bioinformatics, 39(9), p.btad497.

---

### Official Review · Reviewer_w72w · 2026-03-13

**Soundness:** 2
**Presentation:** 3
**Significance:** 3
**Originality:** 3
**Overall Recommendation:** 4
**Confidence:** 4

**Summary:**

this paper addresses the problem of multimodal representation learning with uni-modal pre-trained encoders, and proposes a parameter-efficient fine-tuning approach, for learning low-rank projection sub-spaces to disentangle shared and modality-specific representations. The method combines independence, orthogonality, and mutual information losses to obtain interpretable & disentangled representations from the unimodal frozen encoder representations. Additionally, to overcome challenges of choosing fixed rank, they propose an adaptive rank pruning, that iteratively prunes the learned sub-spaces

**Compliance With Llm Reviewing Policy:**

Affirmed.

**Final Justification:**

The rebuttal meaningfully improves the paper by clarifying important methodological points and by adding results that address concerns raised in my review. In particular, the added missing-modality analysis on real datasets addresses my concern that this setting was motivated in the paper but not evaluated in the original submission. The further clarifications on the method, especially the explanation of the projection design, the handling of the shared representation at inference, and the dimensionality issue in the MI loss, make the technical formulation much clearer.

I am therefore increasing my score. That said, this increase is contingent on these clarifications and the added experimental discussion being incorporated explicitly into the revised paper. In particular, important practical details such as the dimensionality handling in Eq. 7 should appear in the main text rather than only in the rebuttal.

**Key Questions For Authors:**

Q1. The introduction frames data scarcity and missing modality robustness as primary motivations. Can the authors provide experiments evaluating the method under limited data (few-shot) and with missing modalities at inference time?

Q2. Equation 1 defines $z_s = R_s\Phi_1 = R_s\Phi_2$​ as an equality, but exact equality is not enforced. How is $z_s$​ obtained in practice at evaluation time? If $R_s\Phi_1 \neq R_s\Phi_2$​, which projection is used as the shared representation in downstream experiments?

Q3. In Equation 7, the inner product appears dimensionally inconsistent. Please clarify this inconsistency or provide correct form of dimensionalities.

Q4. What is the clear intuitive explanation of why the specific low-rank projection design in Equations 1-3 (particularly the edit function) promotes disentanglement, and how the architectural choices interact with the loss terms to produce the desired subspace structure?

Q5. The ablation in Appendix A.2 claims that removing the MI loss causes performance to drop on both linear-probe and few-shot evaluations, but Table 4 contains no few-shot results. Either provide the missing few-shot experiments or correct the claim.

**Limitations:**

The authors discuss potential applicability to more than two modalities, however argue that it poses challenges on interpretability when more than two modalities are involved. Additionally, they discuss that the performance is bounded by the quality of the unimodal pre-trained encoders.

**Strengths And Weaknesses:**

# Strengths
1. The method is well-suited for domains where interpretability and representational structure matter as much as accuracy. The explicit separation of shared and modality-specific subspaces provides insight into each modality's contribution, which is a meaningful practical advantage over other naive fusion approaches.

2. Paper addresses a meaningful problem and connects parameter-efficient fine-tuning with disentangled multimodal representation learning in a natural and coherent way. Framing multimodal adaptation as a structured low-rank intervention on frozen unimodal encoders is intuitive and principled approach.

3. Experiments on synthetically generated provide rigorous decoupling validation: The use of simulated datasets with known generative structure is a strong methodological choice. By evaluating decoupling against ground-truth latent factors, the synthetic experiments provide a level of validation that real-world benchmarks alone cannot offer.

4. The proposed adaptive rank pruning mechanism is a practical and well-motivated contribution. Table 5 provides concrete evidence that the method reliably converges to compact subspaces. for instance, reducing from 700 to around <40 dimensions on real datasets.

# weaknesses
1. The introduction explicitly motivates the method by (i) the scarcity of paired multimodal data, and (ii) the ability to handle missing modalities. Both are presented as justifications for the framework's design, yet neither are tested in any experiment. All experiments use standard full-data fine-tuning with no few-shot or variable dataset size evaluation, and no experiment with missing modality at inference.

2. Eq 1 defines the shared component as the equality $z_s=R_s\Phi_1=R_s \Phi_2$​, implying that projecting either modality into the shared subspace should yield the same representation. However, this equality is not enforced explicitly. $z_s$​ is obtained at evaluation time: if $R_s\Phi_1 \neq R_s\Phi_2$​ in practice, which projection is used as the shared representation in downstream experiments?

3. Notation inconsistency: The notation switches between $z_s$ and $z_{s1},z_{s2}$ , and between $f_{s},f_{m}$ and $f_{s1},f_{s2},f_{m1},f_{m2}$.

4. there appears to be a dimensionality mismatch between $h_i$ (d-dimension) and $z^{(i)}$ in eq 7: the projection matrices $R_s,R_{m1},R_{m2} \in \mathbb{R}^{r \times d}$ map from d to r dimensions, so both the modality-specific projection $z_{mi}=R_{mi}\Phi_i$​ and the shared projection $R_s\Phi_i$ are r-dimensional. Their concatenation $z^{(i)}$ is therefore 2r-dimensional, while $h_i \in \mathbb{R}^d$. The inner product in Equation 7 is only valid if 2r=d, which is not stated or enforced.

5. While the experiments support effective decoupling between shared and modality-specific representations, the specific design of the projection matrices given in eq 1-3 is not well-justified. It is unclear why projecting fine-tuned representations $\Phi$ into the low-rank subspaces is the right choice, or how the edit structure $f(h) - Rh$ in Equations 2-3 intuitively promotes disentanglement. The paper presents the architecture and loss functions largely in isolation from one another, without explaining how they yield the desired subspace structure. A clearer intuitive explanation of how the projection design interacts with the loss terms to produce disentangled representations would substantially improve the technical clarity of the method.

6. The ablation discussion in Appendix A.2 claims that removing the mutual information loss causes performance to drop on both linear-probe and "few-shot evaluations" across all subspaces. However, Table 4 provides no few-shot evaluation. The few-shot part of this claim is therefore unsupported by the provided evidence, and either the claim should be corrected or the corresponding experiments should be included.

7. minor presentation issues:
- Figure 1 seems to contain mislabeling: the positions of $R_{m2}^T$ and $R_{m1}^T$ seems to be swapped.

---

> ### Author Rebuttal · Authors · 2026-03-30
>
> Thank you for your thoughtful and constructive feedback. We appreciate your recognition of the paper’s practical motivation, the interpretability benefits, the rigor of the synthetic validation, and the usefulness of the adaptive rank-pruning mechanism, and we will use your comments to further strengthen the paper. Below, we address your points in detail.
>
>
> **MultiLoReFT and missing modality:**
> Thank you for raising this important point. Please see our response to Reviewer 4XEn for additional results on this.
>
> **Few-shot performance:**
> We appreciate the reviewer bringing this up. We first would like to highlight that the fine-tuning task on Flickr30 was indeed a few-shot setup. We only used 1K out of the 30K samples for fine-tuning. We realize that this was not adequately communicated in the paper and will make sure to add this explanation explicitly. In response to the reviewer's concerns, we would also like to include more experiments for the few-shot setting to show the robustness of MultiLoReFT. Below are results for Crema-D fine tuning on 1K samples. In general, performance is relatively consistent. However, we observe a slightly larger information leakage in the shared space for sentence ID.
>
> | |Sentence ID|Race|Age|Sex|
> |---|---:|---:|---:|---:|
> |1|82.4±3.7|57.2±3.5|131.1±6.3|86.5±0.9|
> |2|25.0±0.8|64.4±3.7|82.2±20.8|78.1±2.8|
> |3|99.3±0.3|48.5±0.4|153.3±7.5|70.0±4.8|
>
> **Enforcing shared information:** The equality is implicitly enforced by the cross-modal mutual information loss, where the shared representation from the other modality is used to maximize the MI lower bound. This enforces the shared space to include all  information that is relevant to M1 that can be extracted from M2 and vice versa. Also, the losses combined with the pruning remove all redundancy from the representation components, further reinforcing this. During inference, if both modalities exist, we use the average of the 2 shared representations as the shared component.
>
>
> **Intuition for projection design:** Equations 2–3 adopt a low-rank update structure similar to LoReFT, but the update itself is not what enforces disentanglement. Rather, it defines a set of learnable low-rank subspaces to which different semantic roles can be assigned. The disentanglement emerges from the combination of this projection design with the training objectives. Specifically, the orthogonality and independence losses encourage the projected components to capture complementary rather than overlapping information, thereby separating shared and modality-specific factors. The cross-modal mutual information term further constrains the model to preserve informative content while encouraging the shared representation to encode information that is useful across modalities, since it is used interchangeably between them. In addition, pruning imposes a further constraint by removing redundant directions from each subspace, which strengthens the separation and compactness of the learned representations. We will revise the method section to better explain how the projection structure and loss terms work together to produce disentangled representations.
>
>
> **Notation:** We would like to provide some justification for the notation. $z_s$ represents the true underlying shared representation and $z_{s1}$ and $z_{s2}$ are the shared representations approximated by each modality (ideally we would like all these values to be the same). As for the $f_s$ and $f_m$, whenever we don’t use any index, we mean it can be either 1 or 2. We will make sure this is clarified in the text. Also, thank you for pointing out the typo in the Figure.
>
> **Dimensionality in MI loss:** We explicitly handle the dimensionality mismatch in Eq. 7 in our implementation of the MI loss. This can happen especially after pruning changes the dimensionality of the learned subspaces. We agree that this should be stated more clearly in the manuscript and will make sure to add the following detail. As a result, the representations entering the MI loss do not necessarily have matching dimensionality. To compute the MI objective consistently, we therefore project both $h$ and $z$ into a common space using a fixed, non-trainable random projection head, with dimension set to max⁡(dim⁡(h),dim⁡(z))max(dim(h),dim(z)). The MI loss and inner product is then computed in this shared space after normalization. We also evaluated simpler alternatives such as padding or truncation, but found them to be less stable, particularly when pruning substantially changes the effective rank.
>
> **Ablation:** The ablation results show that without the MI loss, information is lost in the projection, as shown by the drop in the downstream prediction performance (linear-probe). We expect that a degradation on the full dataset would likely also hurt few-shot performance, since both depend on the quality of the learned representation. However, we can remove the “few-shot” term from the ablation section.

---

> > ### Author Rebuttal · Reviewer_w72w · 2026-04-04
> >
> > Thank you for the clarifications. The rebuttal addresses several of my concerns and questions, especially the explanation of the projection design and the implementation detail behind the MI loss. That said, these clarifications should be incorporated into the main text. In particular, the fixed non-trainable random projection used before the MI objective is not a minor detail; it is the practical mechanism that makes Eq. 7 well-defined under dimensional mismatch. Without such details, important practical information is missing. The missing clarifications must be made explicit in the paper.
> >
> > The missing-modality concern is partially addressed by the added results, but this evidence is still limited and does not yet cover real-data settings. Overall, I view this concern as only partially resolved.

---

> > > ### Author Response · Authors · 2026-04-07
> > >
> > > Thank you for reviewing our rebuttal in detail, and we are glad that it has been helpful in clarifying your concerns. We will make sure the clarification regarding MI loss is incorporated clearly into the text.
> > >
> > > Regarding the missing modality, as we mentioned in our rebuttal, we are working on expanding the simulated results to all datasets and will incorporate the results in the paper. Here, we would like to show the results expanding the missingness experiments to URFunny dataset (humor detection) and CremaD dataset (emotion detection). We observe similar patterns to simulated settings where fine-tuned representations improve performance in case of missingness. Overall, we envision including these results in the form of bar plots that make comparison much easier. We hope these supplemental results are helpful to resolve the reviewer’s partially resolved concern.
> > >
> > > |                | **URFunny** |        | **CremaD** |        |
> > > |----------------|------------|--------|------------|--------|
> > > |                | H      | $\phi$ |  H      | $\phi$|
> > > | **M2 missing** | 52.2 ± 0.0 | 53.5 ± 0.2 | 59.5 ± 0.0 | 60.9 ± 0.3 |
> > > | **M1 missing** | 60.2 ± 0.0 | 61.0 ± 0.4 | 62.2 ± 0.0 | 63.9 ± 0.7 |

---

### Official Review · Reviewer_EQcL · 2026-03-13

**Soundness:** 3
**Presentation:** 3
**Significance:** 3
**Originality:** 3
**Overall Recommendation:** 4
**Confidence:** 3

**Summary:**

This paper introduces MultiLoReFT, a novel framework that extends low-rank representation fine-tuning (LoReFT) to multimodal learning by decoupling shared and modality-specific subspaces via structured low-rank projection matrices. The framework freezes pretrained encoders and introduces minimal additional parameters while achieving explicit subspace separation, improving both interpretability and downstream task performance. Experiments on simulated and real-world datasets demonstrate competitive generalization and robustness over strong baselines. Despite its contributions, concerns remain regarding scalability beyond bimodal settings and sensitivity to encoder quality.

**Compliance With Llm Reviewing Policy:**

Affirmed.

**Final Justification:**

MultiLoReFT presents a well-motivated, parameter-efficient framework for multimodal representation learning with clear novelty in subspace disentanglement. The rebuttal addressed my main concerns satisfactorily: the trimodal CMU-MOSI experiment demonstrates promising generalizability, and the encoder sensitivity analysis in Appendix A.5 provides useful context. While some clarifications still need to be incorporated into the main text, I am satisfied with the overall direction and maintain my Weak Accept recommendation.

**Key Questions For Authors:**

Please refer to the Limitations Section.

**Limitations:**

**L1: Restricted to Bimodal Fusion**
The current architecture is fundamentally constrained to two-modality inputs, as the subspace decomposition and low-rank projection design do not generalize straightforwardly to settings with more than two modalities. Real-world multimodal applications—such as audio-visual-text fusion in medical diagnosis or embodied AI—routinely involve three or more modalities. Without a principled extension strategy (e.g., hierarchical pairwise decomposition or modality-agnostic subspace aggregation), the practical scope of MultiLoReFT remains narrow. The authors should discuss, at minimum, the theoretical feasibility of such extensions and identify the key design challenges involved.

**L2: Vulnerability to Low-Quality Pretrained Encoders**
MultiLoReFT's performance is fundamentally bounded by the representational quality of the frozen unimodal encoders. When encoders fail to capture task-relevant features—as is common in specialized domains such as rare disease detection, low-resource languages, or degraded sensor data—the low-rank fine-tuning operates on an already impoverished feature space, with no internal mechanism to detect or compensate for such deficiencies. The paper does not analyze encoder quality sensitivity, nor does it propose fallback strategies (e.g., partial encoder unfreezing, quality-aware subspace reweighting, or auxiliary reconstruction objectives). This represents a meaningful robustness gap that should be addressed empirically or at least discussed as a concrete future direction.

**Strengths And Weaknesses:**

# Strengths
**S1: Novel Extension of Low-Rank Adaptation to Multimodal Learning**
MultiLoReFT is the first to adapt LoRA/LoReFT to multimodal scenarios, offering a principled solution to modality information entanglement through structured subspace decomposition.

**S2: Parameter Efficiency and Interpretability**
The use of structured low-rank projection matrices with adaptive pruning ensures minimal parameter overhead while enabling quantitative attribution of each modality's contribution.

**S3: Strong Empirical Generalization**
The method consistently outperforms baselines (e.g., DRIM-U, APOLLO) across both simulated and real-world benchmarks, particularly under data-scarce conditions.

# Weaknesses
**W1: Limited Scalability Beyond Bimodal Settings**
The framework is designed exclusively for two-modality fusion, limiting applicability to real-world scenarios requiring three or more modalities.

**W2: Dependency on Pretrained Encoder Quality**
The framework lacks a compensation mechanism for low-quality or task-misaligned encoders, making performance contingent on upstream representational capacity.

---

> ### Author Rebuttal · Authors · 2026-03-30
>
> Thank you for your thoughtful and constructive feedback. We appreciate your recognition of MultiLoReFT’s novelty, parameter efficiency, interpretability, and strong empirical performance, and we will use your comments to further strengthen the paper. Below, we address your points in detail.
>
>
> **Scalability beyond 2 modalities:**
>  MultiLoReFT is not fundamentally limited to two modalities and the same core formulation extends naturally to larger multimodal settings, as noted in the paper. The same principle applies in more than 2 modalities: the shared subspace is defined as the information shared across all modalities, while each modality also retains its own modality-specific subspace. The same losses extend accordingly: orthogonality and independence are imposed between the shared and modality-specific subspaces, and the mutual-information objective is used in the same way to preserve informative content while encouraging alignment of the shared representation across modalities.
>
> Our current study focuses on the bimodal setting because it allows a cleaner and more rigorous evaluation of disentanglement. To address the reviewer’s concern more directly, we will add an additional experiment on CMU-MOSI using all three modalities (video, audio, and text), for sentiment detection. Because MOSI does not provide ground-truth labels for shared and modality-specific factors, we will evaluate the quality of the learned representation through downstream multimodal prediction performance (Accuracy in binary classification), alongside the same baselines used elsewhere in the paper. For the rebuttal, we use the pretrained features released with the dataset, and simple logistic regression for the sentiment classification. For this rebuttal, we use the pretrained features released with the dataset together with a simple logistic-regression classifier for sentiment prediction. At this stage the resulting performance is therefore not directly comparable to task-specific SOTA models trained on raw inputs with stronger multimodal architectures. Our goal in this experiment is to show that a fully unsupervised MultiLoReFT factorization of the pretrained representations can still improve downstream sentiment prediction in datasets with more than 2 modalitties. In the revised version, we will extend this analysis to stronger pretrained foundation encoders and, where feasible, to raw-input settings as well.
> If one is interested in finer-grained explainability, the framework could be further expanded to include additional subspaces for information shared only between modality pairs. In that case, the orthogonality and independence constraints would also need to be extended to these pairwise-shared subspaces. However, while this is conceptually possible, such a more granular decomposition may not always be useful in practice, and would also make evaluation substantially more difficult.
>
> Late-fusion: 66.90 +- 0.00
> MultiLoReFT: 68.29 +- 0.00
>
>
> **Dependency on Pretrained Encoder Quality:**
>  It is true that MultiLoReFT is a fine-tuning-based method and its performance will depend on the encoder quality. We show this in Appendix A.5 for the Crema-D dataset, that using stronger pre-trained models can significantly boost multimodal performance. This is consistent with many applications that use pre-trained foundation models or few-shot setups[1,2]. The trade-off we want to focus on is when we have small multimodal datasets, and hence limited resources for fine-tuning.
>
> [1] Mañas O, Lopez PR, Ahmadi S, Nematzadeh A, Goyal Y, Agrawal A. Mapl: Parameter-efficient adaptation of unimodal pre-trained models for vision-language few-shot prompting. InProceedings of the 17th Conference of the European Chapter of the Association for Computational Linguistics 2023 May (pp. 2523-2548).
>
> [2] Alayrac JB, Donahue J, Luc P, Miech A, Barr I, Hasson Y, Lenc K, Mensch A, Millican K, Reynolds M, Ring R. Flamingo: a visual language model for few-shot learning. Advances in neural information processing systems. 2022 Dec 6;35:23716-36.

---

> > ### Author Rebuttal · Reviewer_EQcL · 2026-04-06
> >
> > Thank you for the thoughtful rebuttal. Both concerns have been meaningfully addressed.
> >
> > For L1, the trimodal experiment on CMU-MOSI is a welcome addition and demonstrates promising generalizability. I look forward to seeing the extended analysis with stronger encoders in the final revision. A brief formal description of how the orthogonality constraints generalize beyond two modalities would further strengthen this section.
> >
> > For L2, the Appendix A.5 ablation is helpful. I would still appreciate a short discussion in the paper on potential failure cases when no strong pretrained encoder is available, along with possible future directions for mitigation.
> >
> > I am happy to maintain my score of Weak Accept and look forward to the revised version.

---

> > > ### Author Response · Authors · 2026-04-07
> > >
> > > Thank you for your response and your feedback. We are glad that our rebuttal has addressed the reviewer's concerns. We will make sure all the details and clarifications are incorporated into the final version of the paper.

---

### Decision · Program_Chairs · 2026-04-30

**Decision:**

Accept (regular)

**Comment:**

This paper introduces MultiLoReFT, a parameter-efficient framework that leverages low-rank projection matrices to decouple shared and modality-specific information from frozen unimodal encoders. Reviewers appreciated the method's novelty in adapting low-rank interventions to multimodal disentanglement and its strong empirical performance on data-scarce benchmarks. However, initial concerns were raised regarding scalability beyond bimodal settings and a lack of evaluation for missing modalities. I recommend a weak accept because the authors addressed these critiques through new experiments on the trimodal CMU-MOSI dataset and robust missing-modality analysis. The authors must ensure these results and clarifications on mutual information dimensionality are fully integrated into the final manuscript.